# Environmental DNA allows upscaling spatial patterns of biodiversity in freshwater ecosystems

Luca Carraro [1,2 ✉], Elvira Mächler[1,2], Remo Wüthrich[1,2,3] & Florian Altermatt [1,2,4 ✉]

The alarming declines of freshwater biodiversity call for efficient biomonitoring at fine spatiotemporal scales, such that conservation measures be grounded upon accurate biodiversity data. Here, we show that combining environmental DNA (eDNA) extracted from stream water samples with models based on hydrological first principles allows upscaling biodiversity estimates for aquatic insects at very high spatial resolution. Our model decouples the diverse upstream contributions to the eDNA data, enabling the reconstruction of taxa distribution patterns. Across a 740-km$^2$ basin, we obtain a space-filling biodiversity prediction at a grain size resolution of 1-km long stream sections. The model's accuracy in matching direct observations of aquatic insects' local occurrence ranges between 57–100%. Our results demonstrate how eDNA can be used for high-resolution biodiversity assessments in rivers with minimal prior knowledge of the system. Our approach allows identification of biodiversity hotspots that could be otherwise overlooked, enabling implementation of focused conservation strategies.

[1] Department of Evolutionary Biology and Environmental Studies, University of Zurich, Winterthurerstr. 190, CH-8057 Zürich, Switzerland. [2] Department of Aquatic Ecology, Eawag, Swiss Federal Institute of Aquatic Science and Technology, Überlandstrasse 133, CH-8600 Dübendorf, Switzerland. [3] gutwasser GmbH, Geerenweg 2, CH-8048 Zürich, Switzerland. [4] University Research Priority Programme (URPP) on Global Change and Biodiversity, University of Zurich, Winterthurerstr. 190, CH-8057 Zürich, Switzerland. ✉email: luca.carraro@eawag.ch; florian.altermatt@ieu.uzh.ch

Global biodiversity[1,2], and freshwater biodiversity in particular[3–6], are declining at large, unprecedented rates with potentially devastating effects on ecosystems' state and function[2,7], and, subsequently, deleterious consequences for human well-being. To mitigate these threats, effective management and policy making are pivotal, thereby calling for accurate biodiversity data at high spatiotemporal resolution[8–11]. However, most current practices in biomonitoring are still based on localized point-estimates of biodiversity[12–14], which prevents an adequate upscaling at spatially fine scales.

The recent advent of environmental DNA (eDNA, i.e., detection of traces of organisms' DNA in environmental samples) opened new avenues for broadly applicable, fast, efficient, and non-invasive biodiversity assessment in terrestrial, marine and freshwater ecosystems[15–17]. In particular, the use of metabarcoding techniques[18,19] allows parallelized and simultaneous taxonomic identification of many species from a single analyzed eDNA sample. Nonetheless, spatial projections of eDNA data and their quantitative interpretation to derive patterns of species' richness (let alone abundance) are still challenging, which is why eDNA has hitherto mostly been used for local assessments of biodiversity[20–22]. However, knowledge on biodiversity at large scale and high spatial resolution is paramount for effective biodiversity management and identification of regions of high conservational value. Recently, it has been postulated that, owing to the transport of genetic material with stream water, the use of eDNA in rivers conveys information on biodiversity of the upstream catchment[23]. Specifically, the eDNA sampled at a river's site results from the aggregation of locally shed DNA traces and the dynamics of molecules transported[24–26], from a number of upstream sources (i.e., the species' locations) along a dendritic river network towards the sampling site. On the one hand, this feature highlights the potential key role of eDNA as a tool for monitoring biodiversity at large scales[11]; on the other hand, it also poses additional challenges with regards to the reconstruction of spatial patterns of biodiversity at highly resolved scales, because a DNA signal may not always match with a species' local occurrence. Furthermore, eDNA advection is not a sole transportation process only, but is also subject to decay dynamics typically dependent on several abiotic (e.g., temperature, pH, substrate) and biotic (e.g., microbial activity) factors, which can differently impact the fate of eDNA molecules in space and time[24,25,27–29].

Here, we develop an integrated hydrology-based modeling framework (hereafter termed eDITH—eDNA Integrating Transport and Hydrology), built on the approach of refs. [30,31], to analyze metabarcoding eDNA data collected at 61 locations in a 740-km² river basin and derive the spatial patterns of all aquatic insect taxa belonging to the orders of may-, stone-, and caddisflies (Ephemeroptera, Plecoptera, and Trichoptera, abbreviated as EPT). EPT taxa show a high spatial variation in their richness and community composition[14,32,33] and are characterized by a high sensitivity to pollutants, which makes them widely used as indicators of water quality[34]. We then produce spatially finely resolved maps of presence and relative abundance of each taxon at a mean resolution of 1-km long stream segments (2.5th–97.5th percentiles: 0.07–3.18 km), which are validated by comparison with local abundance data obtained via standardized[35] kicknet sampling. The ensemble results enable us to identify the portions and locations of the catchment with highest EPT taxa richness.

## Results

**Data collection and model structure.** Three independently replicated eDNA samples and a pooled benthic invertebrate kicknet sample were taken at each of 61 sites (60 for kicknet) across the Thur catchment in Switzerland (Fig. 1) in June 2016.

Site selection, data collection and technical processing of samples for both eDNA and kicknet are described in the "Methods" section and in Mächler et al.[36], the original publication of the presented sequencing data. In short, amplicon sequencing of a short barcode region of the cytochrome c oxidase I (COI) was run for each of the 183 water samples (three per site) with Illumina MiSeq sequencing platform, allowing us to identify 50 EPT genera, present in at least one replicate at one site. The metabarcoding procedure and subsequent bioinformatic pipeline resulted in 423,043 sequences (i.e., reads) for the detected EPT genera; the median number of reads per site, pooled over genera and replicates, was 3,406; the median number of reads per genus, pooled over sites and replicates, was 1,637. Organisms collected via kicknet belonged to 47 EPT genera, of which 36 were also found via eDNA.

We applied the eDITH modeling framework to reconstruct the spatial distributions of the 50 EPT genera detected by the metabarcoding analysis. Such framework couples: a species distribution model relating target taxa abundance to environmental covariates; dynamics of eDNA shedding from a multiplicity of sources; eDNA advection (and relative decay) along the river network to the sampling site; and a measurement error model that accounts for the uncertainties related to the metabarcoding procedure. To provide a quantitative interpretation of the metabarcoding data, we assumed that read numbers followed a geometric distribution with mean proportional to the (site-dependent) eDNA concentration predicted by our model. By fitting the eDITH model on read number data for the 50 different EPT genera, we were able to assess the role of environmental covariates in driving the spatial distribution of single taxa (Fig. 2a) and EPT biodiversity (Fig. 2b, c) across the study catchment. Moreover, we estimated characteristic decay times of eDNA (mean across all genera: 1.5 h; the distribution of values is shown in Fig. 3). The proposed approach has the ability to transform taxon occurrence estimates from local, point-based measurements into space-filling segments, framing a complete picture of spatial patterns of relative density for the target taxa at a catchment scale. In order to facilitate the comparison between model results and kicknet data, relative density maps for all genera obtained by eDITH were converted into maps of detection probability (see examples in Fig. 4), and, in turn, by imposing a threshold of 2/3 on detection probability[36], into presence maps. Subsequently, by summing up all genus presence maps, we evaluated the spatial patterns of genus richness for the study catchment (Fig. 5a).

**Biodiversity predictions.** According to model predictions (Fig. 5a), the central latitude headwaters are richer in EPT genera, while the downstream reaches and one side tributary (Glatt) host a smaller biodiversity. This observation matches geographical covariates for this latter portion of the catchment (TH1, TH2, GL1, GL2—see Fig. 1b) having a significant negative effect on many genera (Fig. 2c) and the higher pollution levels that characterize the Glatt[37]. In particular, eDITH enables the identification of biodiversity hotspots (see ovals in Fig. 5a) that were not captured by either eDNA data (Fig. 5b) or kicknet sampling (Fig. 5c) alone. Importantly, such a finding is not merely an artifact of model extrapolation: in fact, our model's predictions are based on a number of data points from sites downstream of these hotspots, which embed information on the upstream regions.

As shown in Fig. 6, the distribution of genus richness predicted by the model in the headwaters (reaches of stream order 1) matched the one assessed via the kicknet sampling: a two-sample Kolmogorov-Smirnov (2KS) test did not reject the null hypothesis

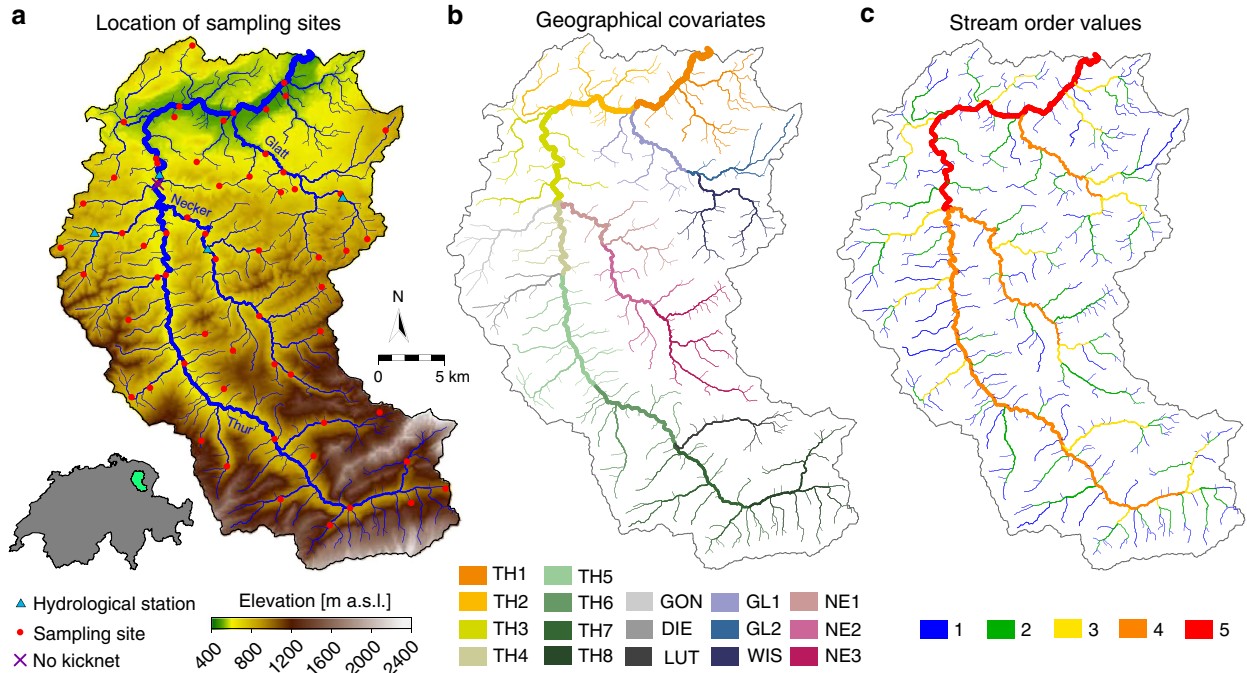

**Fig. 1 Overview of the study area. a** Digital elevation model of the Thur catchment and the respective extracted river network used in the analysis. eDNA and kicknet sampling sites are identified by circles. Kicknet was not performed in one site (denoted by a cross). In the bottom-left insert, the location of the Thur catchment within Switzerland is shown. The blue triangles show the positions of the hydrological stations. **b** Partition of the river network into 17 clusters, corresponding to the respective geographical covariates. The legend indicates the covariates' ID, with letters abbreviating the name of the corresponding stream (TH = Thur, GON = Gonzenbach, DIE = Dietfurterbach, LUT = Luteren, GL = Glatt, WIS = Wissbach, NE = Necker). **c** Strahler stream order values for all network reaches. Note that stream order is calculated on the extracted river network; as a result, stream orders differ from those used by Mächler et al.[36].

that the two samples come from the same distribution ($p = 0.43$). Conversely, eDNA data alone underestimated genus richness in low stream order reaches; a 2KS test between eDNA and kicknet richness values at reaches with stream order 1 yielded $p < 0.001$. This is presumably due to the low number of replicates per single location and the high heterogeneity among these. Instead, the EPT biodiversity predicted by the model at the high ($\geq 4$) stream order reaches was lower than the one measured by the kicknet dataset (2KS test: $p = 0.002$), whereas in these reaches the genus richness based on eDNA data matched the kicknet-based richness (2KS test: $p = 0.98$). Such results can be attributed to the fact that, unlike the biodiversity assessments performed via kicknet and eDNA data, model predictions are based on the ensemble of sampling sites, thereby information on a given headwater reach is potentially contained in all the samples taken at any location downstream from the headwater in question. On the contrary, model predictions at the downstream portion of the catchment are solely based on information from downstream sampling sites, and are thereby more prone to error, since the model might interpret an eDNA signal detected at a downstream site as originated from an upstream source rather than locally.

**Goodness of fit and accuracy of model predictions.** A goodness-of-fit test (Fig. 7a) showed that the model adequately reproduced the observed patterns of read numbers. In particular, we formulated the null hypothesis $H_0$ that, for each genus, observed read numbers come from the hypothesized geometric distribution with mean predicted by our model. Remarkably, for all genera, in >90% of all sites $H_0$ could not be rejected, indicating a very high goodness of fit.

The accuracy of genus presence maps produced by our approach was assessed via comparison with the kicknet data.

Accuracy was evaluated as the fraction of sites where the presence or absence predicted by the model matched the kicknet observation. The average accuracy across all genera was 82.4% (range: 40–100%; see Fig. 7b). If also false positives, that is, sites where the model predicted presence but kicknet assessed absence, were considered as plausible model predictions (which is likely the case for elusive genera), average accuracy increased to 92.8% (range: 56.7–100%).

**Comparing genus distribution maps with faunistic knowledge.** Remarkably, a fair agreement can be observed between the eDITH-based predictions of spatial distribution of taxa (Fig. 4), the related predicted role of covariates (Fig. 2a) and the ecological and faunistic knowledge on taxa[38]. We here discuss the results obtained for three representative taxa. Only two species belonging to the mayfly genus *Habroleptoides* (*H. confusa* and *H. auberti*) are known to occur in Switzerland and in the Thur catchment. *Habroleptoides auberti* is reported in the upper Thur and Necker basins, which largely reproduces the patterns displayed in Fig. 4a, d. *Habroleptoides confusa* is reported to occur in waters of neutral pH, which may explain the negative relation of swamps (covariate L-SW) and peat (G-PE) found by the model (see line 31 of Fig. 2a). Both species are sensitive to pesticides, which justifies the predicted negative role of urban areas (L-UR), drainage area (L-DA) and orchards (L-OR). As for the stonefly *Protonemura*, three species (*P. brevistyla*, *P. auberti*, and *P. nitida*) are expected to be present in larval stages in the Thur catchment in late June (when the sampling was performed), while the emergence of *P. lateralis* is likely to occur earlier in the season. All three species are commonly found in spring brooks, epirhithral and metarhithral (i.e., upper and middle upland) streams, but also in unpolluted waterbodies. This is in agreement with the predicted

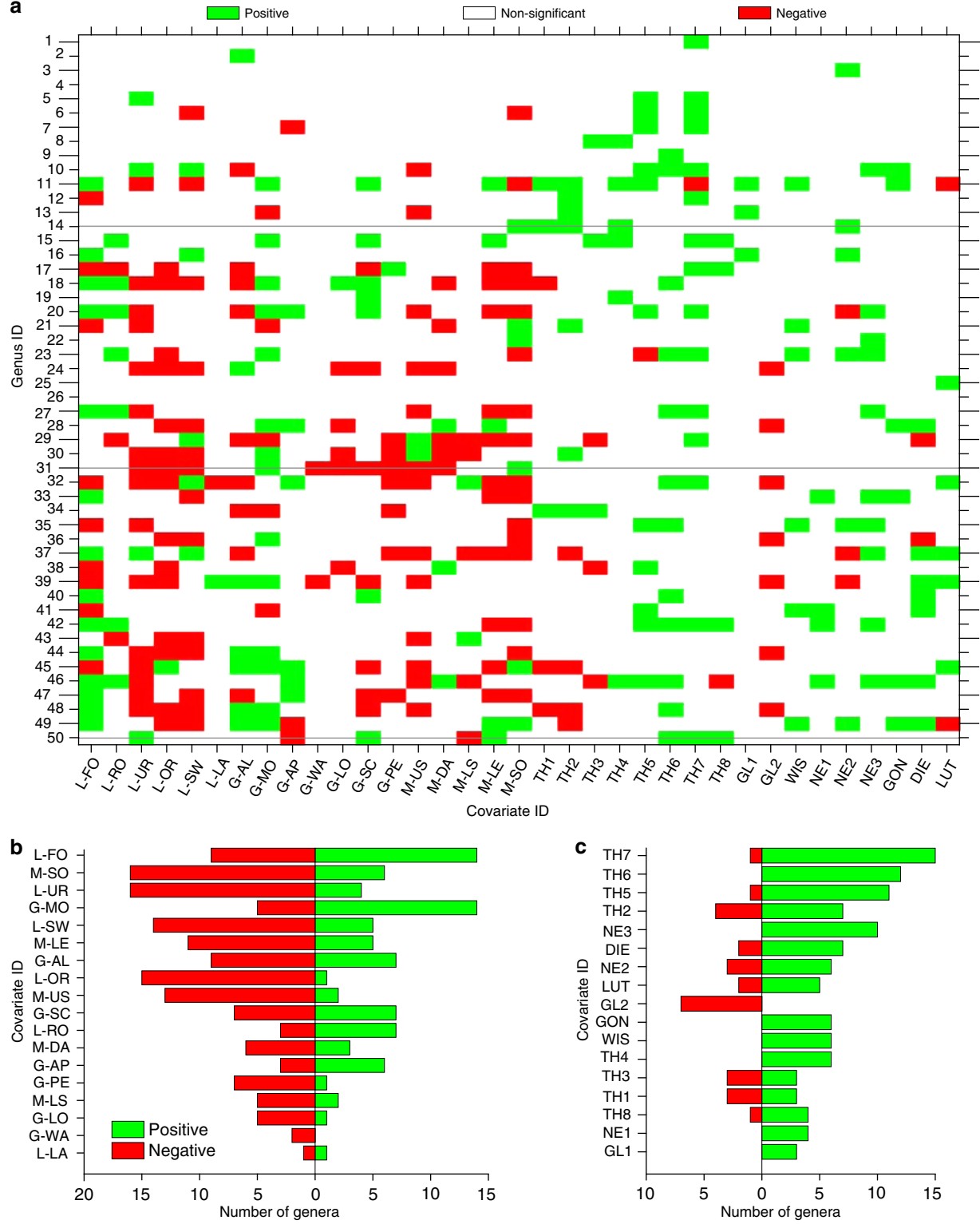

**Fig. 2 Relationship between covariates and genus density. a** Effect of covariates on predicted genus density. Genus IDs are as in Supplementary Data 1, covariate IDs are as in Supplementary Table 1. A covariate effect is deemed positive (negative) if the 95% equal-tailed credible interval of the posterior distribution of the corresponding $\beta$ parameter is above (below) zero. Non-significant effects are those whose 95% equal-tailed credible interval of the posterior distribution of the corresponding $\beta$ parameter encompasses zero. Horizontal lines identify the genera whose distribution maps are shown in Fig. 4. Bottom row: aggregate effect of morphological, geological, land cover (panel **b**), and geographical (panel **c**) covariates on all genera. The x-axis displays the number of genera for which a given covariate has a significant positive or negative effect.

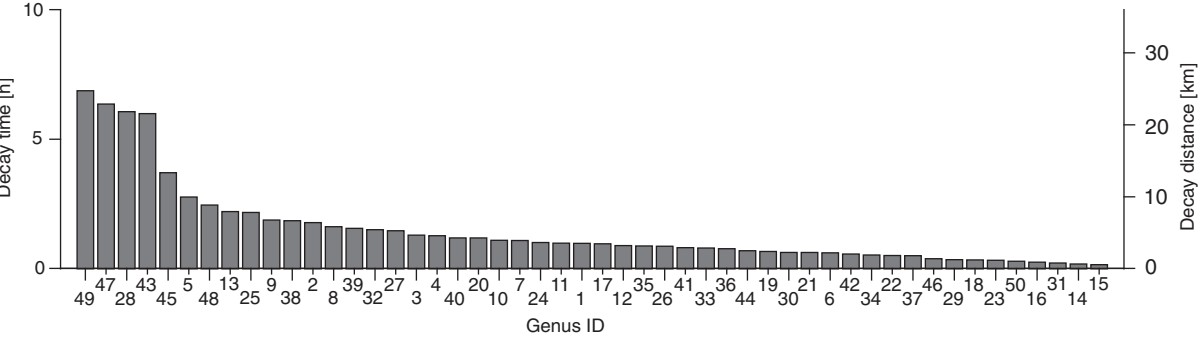

**Fig. 3 Estimates of decay times.** Values displayed correspond to the medians of the sampled posterior distributions. Genus IDs are as in Supplementary Data 1. The right *y*-axis converts decay times into distances, by assuming a constant water velocity equal to 1 ms$^{-1}$. Source data are provided as a Source Data file.

*Habroleptoides*
(Ephemeroptera)

*Protonemura*
(Plecoptera)

*Athripsodes*
(Trichoptera)

**Fig. 4 Examples of relative density maps (top row) and detection probability (bottom row) for three genera belonging to the three different orders.** Values shown correspond to medians of the sampled posterior distributions of $kp_i$ (top row) and their equivalent in terms of detection probability according to Eq. (6) (bottom row). Source data are provided as a Source Data file.

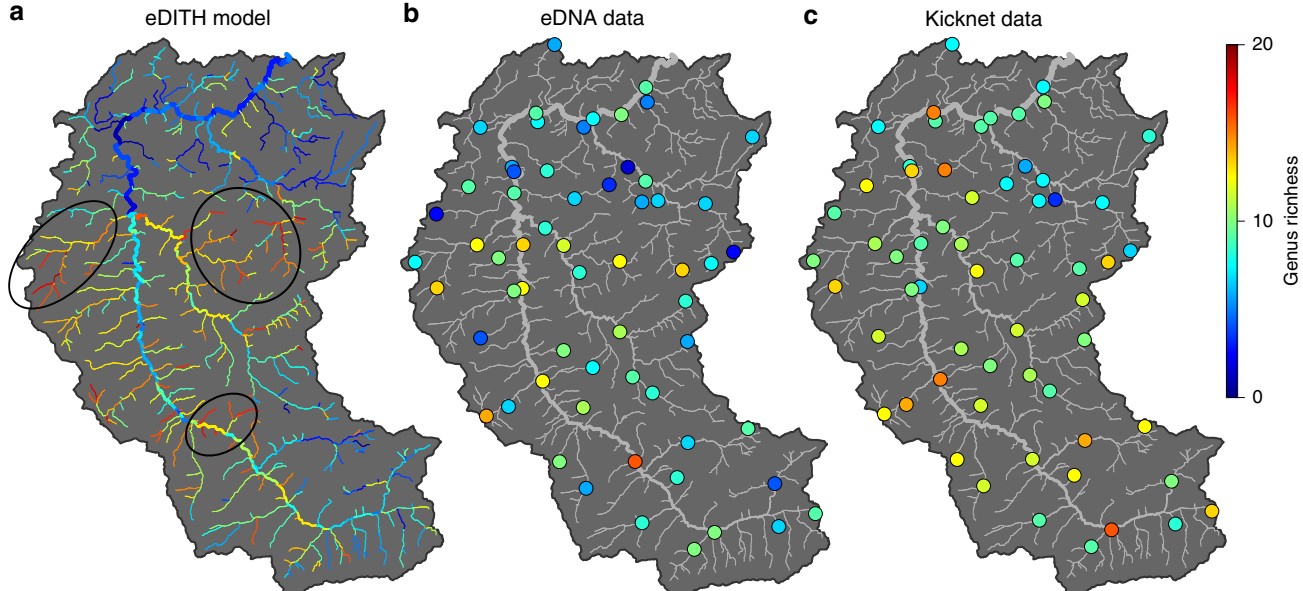

**Fig. 5 Genus richness maps and accuracy of eDITH predictions. a** Genus richness map predicted by the eDITH model. Black ovals identify possible biodiversity hotspots highlighted by model results. **b** Genus richness from eDNA data; here presence of a genus at a site is attributed if at least 2 out of 3 replicates have nonzero read numbers[36]. **c** Genus richness obtained by kicknet sampling. Source data are provided as a Source Data file.

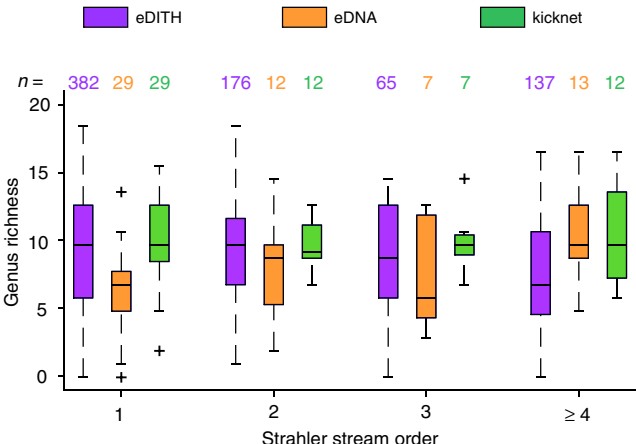

**Fig. 6 Distribution of genus richness as evaluated by the eDITH model, the eDNA data, and the kicknet sampling as a function of stream order.** Central lines represent medians; bottom and top boxes' edges represent 25th and 75th percentiles, respectively; whiskers span the data range, up to 1.5 times the interquantile range; data points exceeding such threshold are considered outliers (marked with crosses). Genus richness values are as in Fig. 5. Sample sizes $n =$ numbers of reaches/sites for each column of the boxplot are also reported. Stream order values across the river network are displayed in Fig. 1c. Source data are provided as a Source Data file.

distribution maps shown in Fig. 4b, e and with the positive role found for scree (G-SC, see line 50 of Fig. 2a), local elevation (M-LE) and the upper geographical clusters of the Thur (TH6, TH7, TH8—compare with Fig. 1b). Finally, the stonefly *Athripsodes* is mostly represented by the species *A. albifrons* in the study catchment. This eurythermal species is typically found in lower rhithral and epipotamal reaches, characterized by midsize to large river widths and warm summer temperatures. The predicted distribution maps for *Athripsodes* (Fig. 4c, f) and the positive role found for geographical clusters TH1, TH2, TH4 and NE2 (see line 14 of Figs. 2a and 1b) match such empirical observations, although model results predict absence of *Athripsodes* in clusters TH3 and NE1, which are also plausible habitats for this taxon.

**Cross-validation analysis of the effect of sample size**. To corroborate our results, we also performed a cross-validation study, where additional model simulations (AS) were trained on subsets of the eDNA sampling sites and their performance was compared to the one of the original, complete model (CM—see "Methods" section). Genus-specific results are displayed in Fig. 7c, d, while aggregated results are reported in Table 1. Notably, the predictive performance of the model is only minimally affected by the reduction in sampling points: simulations trained on only 40% of the sites had an average loss of goodness of fit and accuracy of 6.55% and 4.07%, respectively. Expectedly, the loss of goodness of fit of models trained on a limited set of sites is mainly to be attributed to validation sites, whereas the fit on calibration subsets is improved with respect to CM, as indicated by the negative values in the corresponding row of Table 1. This is reasonable, as models trained on small datasets $s$ cannot perform worse than models calibrated on larger datasets $l \ni s$ (i.e., $s$ is a subset of $l$) with respect to $s$. Indeed, if this were not the case, it would imply that the calibration of the model trained on $s$ did not converge.

## Discussion

As widely recognized[11,16,23], the use of eDNA in rivers leads to a faster and less invasive biodiversity monitoring as compared to kicknet sampling, but its results consist in a number of pointwise estimates that could hardly be projected into biodiversity maps at high spatial resolution. Moreover, these estimates are confounded by eDNA advection and can therefore only be used for an aggregated, qualitative biodiversity assessment. Our approach, instead, by coupling spatially distributed eDNA samples with a model based on hydrological first principles, allowed an estimation of the catchment-scale spatial distribution of aquatic insects' presence at a high spatial resolution, which is hardly achievable via traditional sampling techniques[11,14]. This potentially constitutes a game-changer in biodiversity and ecosystem studies, with clear benefits for biomonitoring and conservation programs.

Our framework opens avenues for freshwater ecosystem management, as it allows non-invasive and efficient localization of, for example, endangered or invasive taxa, pathogens, as well as biodiversity assessments over taxonomic groups wider than those

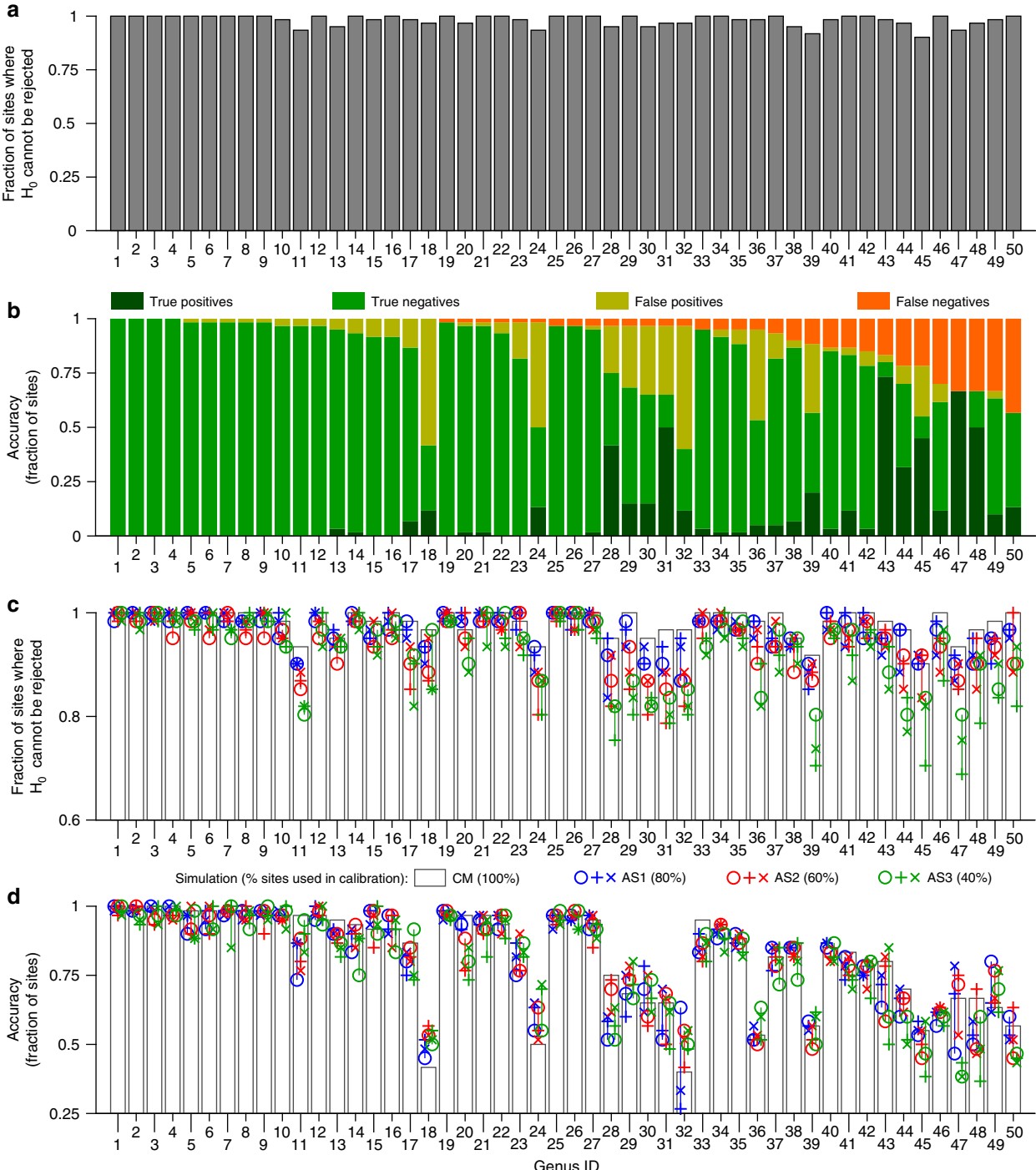

**Fig. 7 Goodness of fit and accuracy of model predictions.** Genus IDs are as in Supplementary Data 1. **a** Goodness-of-fit test for all genera. In the y-axis label, $H_0$ is the null hypothesis that the triplet of read numbers evaluated at a site for a given genus comes from a geometric distribution with mean predicted by Eq. (3). **b** Accuracy of the modeled presence maps with respect to the kicknet sampling for each genus. Genera for which only true negative and false positive sites were identified are those undetected by kicknet sampling. **c**, **d** Goodness of fit (panel **c**) and accuracy (panel **d**) of model results as a function of number of sampling sites used in calibration. White bars in panels **c** and **d** reproduce the same values as in panels **a** and **b**, respectively. Colors identify simulation groups, corresponding to different sizes of calibration subsets. For a given color, each symbol type identifies each particular simulation (corresponding to a given calibration subset). Note that identical symbols with different colors do not indicate any relationships among the respective simulations. Colored vertical lines, spanning ranges of variables for each genus and simulation group, are added for visual purposes. Source data are provided as a Source Data file.

**Table 1 Performance of additional simulation models.**

| Simulation group | AS1 | AS2 | AS3 |
|---|---|---|---|
| Fraction of sites used in calibration | 80% | 60% | 40% |
| Loss of GOF with respect to CM (all sites) | 1.88% | 3.87% | 6.55% |
| Loss of GOF with respect to CM (calibration subsets) | −0.27% | −0.77% | −1.04% |
| Loss of GOF with respect to CM (validation subsets) | 10.67% | 11.03% | 11.81% |
| Loss of accuracy with respect to CM (all sites) | 1.86% | 2.04% | 4.07% |

Aggregated statistics for the performance of additional simulation (AS) models, trained on subsets of eDNA sampling sites, as compared to the complete model (CM). Aggregation is performed by averaging results for all genera and all simulations within a simulation group. Note that the loss of goodness of fit (GOF) on all sites is given by the average of the same variable calculated on calibration and validation subsets, weighted by the sizes of the subsets. Source data are provided as a Source Data file.

here used as case study[11,19,23,39]. Indeed, applying eDITH to motile organisms (such as freshwater fish, crustaceans or mollusks) is possible because the scale distance of hydrological transport in rivers (say, considering a reference water velocity of $1\,\mathrm{m\,s^{-1}}$) is much larger than that of mobility of most organisms[40,41]. In principle, application to terrestrial taxa inhabiting areas with high drainage density of the stream network is also possible. In this case, the source area $A_s$ (see Eq. (1)) should be defined as the subcatchment area, rather than the riverbed area as in the present case. However, further caution in the modeling approach is required, as only a fraction (possibly dependent on the distance from the drainage network) of the eDNA shed by such organisms will be transported downstream by the streamflow.

The eDITH model relies on estimates of relevant hydrological variables such as water velocity and discharge across the catchment. In the case study here presented, we obtained these data from power-law regressions based on four hydrological stations. This approach allows capturing the essential features of hydrological transport and hydrological dynamics at large scales, as studied herein, and is supported by longstanding evidence in hydrology and geomorphology[42–44]. Importantly, all uncertainties associated with the processes of eDNA shedding, transportation, decay, extraction, and bioinformatic processing (all of which are coarsely accounted for in the eDITH approach) are at a scale that a more elaborate hydrological model would not significantly improve the predictions of taxa distribution and biodiversity. The use of universal hydrological relationships also allows the application of our approach to riverine systems with scarce hydrological data, thereby enabling monitoring of highly biodiverse but hardly accessible ecological systems[45].

In our case study, biodiversity predictions produced by eDITH proved to be rather robust to the choice of sampling sites, as highlighted by our cross-validation analysis (see Fig. 7c, d and Table 1). However, the subsamples of sites used in such investigation were chosen such that the proportions of sites from the upstream and downstream portions of the catchment (reflected by the stream order values of the corresponding reaches) were not altered with respect to the original set of sites. We also observed that including information from downstream sampling sites in the eDITH model is instrumental in evaluating taxon abundance in the upstream reaches (due to eDNA transport), but generally leads to poor prediction of taxon abundance at local, downstream communities (see Fig. 6). We therefore suggest using our model to assess biodiversity in the portion of the catchment located some kilometers upstream of the most downstream eDNA sampling site. Measurements taken at this site are still needed for estimation of the upstream pattern of taxon distribution, but will not be informative enough to disentangle the contribution to the eDNA signal from the nearest sources. Moreover, further studies[46] are needed to investigate how the positioning of eDNA sampling sites within a catchment influences the prediction power of the eDITH model.

A key challenge towards a quantitative use of metabarcoding data is expressing the relationship between number of reads and the underlying eDNA concentration for a given taxon. Although several studies (e.g., refs. [47–49]) found that high numbers of reads are generally related to high abundances/biomass of species, which is potentially reflected in high eDNA concentrations, a deterministic relationship cannot be found, due to the high stochasticity of read number values resulting from the uncertainties of multiple steps of the eDNA laboratory procedures. For example, different extraction methods can influence diversity results (e.g., refs. [50,51]), sequencing platforms have specific error rates when generating DNA sequences (see ref. [52]), and primer bias can lead to distorted abundance proportions (e.g., refs. [53–55]). However, it has to be noted that the efficiency of the primer used herein is relatively similar among the three inspected insect orders: an in-silico evaluation of primer performance showed that the efficiencies of the forward primer for Ephemeroptera, Plecoptera and Trichoptera are 76%, 77%, and 80% respectively, while those of the reverse primer are 100%, 100%, and 98%, respectively[56]. We expect that recently developed primers[57] even more optimized for EPT taxa additionally strengthen the approach proposed here. Furthermore, in our approach the parameter $p_0'$, which transforms relative density distributions—proportional to $\exp(\boldsymbol{\beta}^T \boldsymbol{X}(i))$—into read numbers, was estimated independently for each genus by the calibration procedure. Hence, possible differential affinities between different eDNA sequences do not affect model results.

In this work, we propose the use of a geometric distribution for read number data from the same sample, whose mean is proportional to the eDNA concentration of the sample. Such choice enables exploiting the quantitative information contained in the read number values, while accounting for their large stochasticity. The choice of such distribution was based on the analysis of the data available for this case study (see "Methods" section), but it would benefit from a validation based on a lab study. Therefore, we call for further research to better elucidate this aspect.

Finally, it is worthwhile to note that a quantitative interpretation of eDNA data in rivers is crucial even if the ultimate goal is to make biodiversity predictions based on estimates of presence or absence of the investigated taxa. Indeed, in order to understand eDNA advection and decay dynamics across a river network, it is essential to adopt a mechanistic approach such as eDITH, which frames the problem of eDNA transport in terms of quantities of eDNA shed in the different upstream sources. This enabled us to derive predictions of spatial patterns of relative taxon density (Fig. 4); however, as a conservative assumption, both the ground-truthing of model predictions against the kicknet data and biodiversity estimates were operated on a presence/absence basis.

In summary, our work shows how the combined use of eDNA and hydrology-based modeling allows the upscaling and prediction of aquatic biodiversity across whole river networks at a very fine resolution. Based on empirical data from a few tens of

sampling sites for eDNA, one can reconstruct biodiversity patterns on species richness at resolutions down to 1-km stream lengths segments over hundreds of square kilometers. Our approach thereby opens up the possibility to map biodiversity across riverine systems worldwide. Given that biodiversity in these ecosystems is among the most threatened, such information is crucially needed for protection and conservation measures.

## Methods

**Data collection and sequencing.** In June 2016, diversity data were collected at 61 sites in a 740 km$^2$ sub-catchment of the river Thur, northeastern Switzerland. No singular rain events took place during the sampling days, and thus all sampling was performed at base-flow conditions. Sampling sites were chosen in order to proportionally represent all stream orders in the river network (Supplementary Fig. 1) and to span the complete geographical extent of the catchment. At each site, a standardized[35] 3-min kicknet sampling applied to three microhabitats was performed to collect benthic macroinvertebrates. The subsamples from the three microhabitats were pooled and stored in ethanol. In the laboratory, debris was removed and all individuals belonging to may-, stone-, and caddisflies (Ephemeroptera, Plecoptera, and Trichoptera, abbreviated as EPT) were identified under a stereomicroscope to the genus or species level if applicable. One sample was lost due to handling error, and thus we subsequently only had EPT kicknet sample data from 60 sites.

At the same 61 sites at which kicknet data were collected, also eDNA samples were taken. We collected three independent samples of 250 mL of river water at each site (sampled below the surface and well above the river bottom). In small streams (up to about 1-m wide), the water was taken from the middle of the stream, while in larger streams the water was taken about 0.5 m away from the shore side. All three water samples per site were filtered on site on separate GF/F filters (pore size 0.7 μm Whatman International Ltd.), which were stored on ice immediately after filtering, and frozen at −20 °C within a few hours. Subsequently, these three samples were analyzed separately and as independent replicates. To prevent cross-contamination, the eDNA samples were collected a few meters upstream of the kicknet sampling, and kicknet sampling and eDNA sampling were performed by different people.

DNA was extracted with the DNeasy Blood and Tissue Kit (Qiagen GmbH) from the filter in a lab dedicated to low DNA-concentrated environmental samples (i.e., clean lab with positive air pressure, well-defined procedures and all work conducted under sterile benches, and no handling of high-DNA concentration samples or post-PCR products). A short barcoding region of the COI gene[58,59] was targeted, and a dual-barcoded two-step PCR amplicon sequencing protocol for Illumina MiSeq was performed. In short, three PCR replicates were performed for each eDNA replicate with primers that contained an Illumina adapter-specific tail, a heterogeneity spacer and the amplicon target site. These three PCR replicates per sample replicate were pooled and indexed in a second PCR with the Nextera XT Index Kit v2 (Illumina). We measured the concentration of all indexed PCR reactions and pooled them in equimolar parts to a final library, which we ran twice on two consecutive Illumina MiSeq runs to increase sequencing depth. Raw data were demultiplexed and read quality was checked with FastQC[60]. Thereafter, end-trimming (usearch, version 10.0.240) and merging (Flash, version 1.2.11) of raw reads were performed, before primer sites were removed (cutadapt, version 1.12) and reads were quality-filtered (prinseq-lite, version 0.20.4). Next, we used UNOISE3[61], which has a built-in error correction to reduce the influence of sequencing errors, to determine amplicon sequence variants (ZOTUs). To reduce sequence diversity, we implemented an additional clustering at 99% sequence identity. We checked ZOTUs for stop codons of the invertebrate mitochondrial code, to ensure an intact open reading frame. For the taxonomic assignment, all COI-related sequences were downloaded from NCBI and ZOTUs were blasted against the NCBI COI collection. We extracted the top five best blast hits and then used the R packages "taxize"[62] (version 0.9.7) and "rentrez"[63] (version 1.2.2) to acquire taxonomic labels. We modified the selected COI sequences with the taxonomic labels in order to index the database. The ZOTUs were then assigned to taxa using Sintax (usearch) and the NCBI COI-based reference. Details of the data collection and the bioinformatic pipeline can be further assessed in Mächler et al.[36]. The OTUs found in this dataset are generally saturating[39]. We acknowledge that the primers used are targeting eukaryotic diversity in general and were not only specific to EPT taxa, which might lead to an underestimation of detections of EPT by eDNA (i.e., false absences). This, however, is conservative with respect to our findings, as we would miss taxa with eDNA where the kicknet sample indicated their presence.

**Network extraction and hydrological characterization.** The river network was extracted from a 25-m Digital Elevation Model (DEM) of the region provided by the Swiss Federal Institute of Topography (Swisstopo) by applying the D8 method through a TauDEM routine in a GIS software[64]. A threshold of 800 pixels (0.5 km$^2$) on contributing area was applied to identify the channelized (i.e., perennial, see ref. [64]) portion of the drainage network, whose total length equalled

751 km. We then defined a reach as a sequence of pixels of the DEM matrix, originating from a source or a confluence, directed downstream, and ending at the following confluence or at the outlet. This resulted in a discretization of the river network comprising 760 reaches of median length 0.78 km (95% equal-tailed interval of the distribution: 0.07–3.18 km). Note that a lower threshold area would have resulted in a higher number of reaches, implying a more refined discretization of the river network but also an increased computational burden for the subsequent model runs. Hence, we chose the highest value of threshold area such that: i) the extracted river network retained all reaches where sampling sites were located; ii) a qualitative comparison with the vectorial hydrographic network provided by Swisstopo resulted adequate. For modeling purposes, reaches were considered as nodes of a graph with edges following flow direction. All variables referred to a node were considered homogeneously distributed within the corresponding reach.

In order to assess values of hydrological variables for all reaches, we made use of power-law scaling relationships, a well-established and universally applicable concept in hydrology[42–44]. In particular, river width $w$, river depth $D$ and water velocity $v$ are known to scale (both within a single cross-section and in the downstream direction) as a power-law of water discharge $Q$ (refs. [42,43]); along the flow direction, the relationships $w \sim Q^{0.5}$, $D \sim Q^{0.4}$, $w \sim Q^{0.1}$ are valid over wide ranges of natural streams[42]. Moreover, water discharge scales linearly across a catchment with drainage area $A$ (ref. [44]). Strictly speaking, the relationship $Q \sim A$ holds for mean annual values of $Q$; however, it can be reasonably extended to values of $Q$ averaged over shorter time windows (say, at least 1 day), provided that the time scale of flow propagation is much shorter than 1 day, and that rainfall (and the resulting runoff generation) can be considered spatially homogeneous if aggregated at a daily scale. Both assumptions are reasonable for catchments up to $10^3$ km$^2$ (ref. [65]), as the one here studied. Mean water discharges during the sampling days and stage-discharge relationships were available at four stations operated by the Swiss Federal Office for the Environment (FOEN) (see Fig. 1). River widths at these locations were estimated via aerial images. Power law relationships with drainage area for discharge and river width were fitted on the four stations, yielding $Q = 0.072 A^{1.056}$ and $w = 1.586 A^{0.526}$, where $A$ is in km$^2$, $Q$ in m$^3$s$^{-1}$, and $w$ in m. As for river depth, we discarded the station with lowest drainage area because we observed that the values of depth measured therein were highly overestimated with respect to the expected values, based on the other stations and the scaling exponent 0.4 (ref. [42]). Hence, we limited the fit to the three other stations and obtained $D = 0.073 A^{0.463}$, where $D$ is in m. By assuming rectangular river's cross-sections, we finally derived a power-law relationship linking water velocity $v$ to drainage area: $v = Q/(Dw) = 0.623 A^{0.067}$, where $v$ is in ms$^{-1}$. Notably, all scaling exponents obtained were very close to the literature values[42,44]. Details on the fit of these hydrological relationships are reported in Supplementary Fig. 2.

**Choice of covariates.** A first set of 18 covariates was chosen as representative of morphological, geological and land cover features of the catchment. These covariates can either reflect local or upstream characteristics. Land cover covariates were evaluated as local values, because they are assumed to potentially have a role in determining local taxon suitability (see also ref. [66]). Geological covariates were calculated as upstream averaged values, because they are likely to affect the chemical composition of streamflow at a site; such process could affect local habitat suitability but is driven by the upstream catchment[30]. Morphological covariates were obtained by analyzing the DEM of the region, while geological and land cover raster maps were provided by Swisstopo[67,68]. The list of covariates is shown in Supplementary Table 1.

A second set of covariates was constituted by grouping all 760 reaches into 17 clusters according to geographical proximity and hydrological connectivity (see Fig. 1c). For each cluster, a corresponding covariate vector was defined with values equal to one for the reaches constituting that cluster, and zero otherwise. The addition of these 'geographical' covariates aimed at allowing eDITH to reproduce spatial patterns uncorrelated to any of the previous covariates (e.g., in the case of a taxon only inhabiting a single tributary of the catchment).

The 35 covariates were z-normalized and checked for possible multicollinearity. The 760-by-34 design matrix obtained by removing the covariate corresponding to cluster LUT of Fig. 1c (corresponding to the Luteren tributary, and arbitrarily chosen among the geographical covariates: each of them can indeed be expressed as a linear combination of the other 16 geographical covariates) yielded all pairwise correlation coefficients with absolute values lower than 0.80 and variance inflation factors lower than 10, which minimized the effects of multicollinearity[69]. The LUT covariate was however kept in the design matrix to facilitate model fitting.

**The eDITH model.** The eDNA transport component of the eDITH model is derived from Carraro et al.[31]

$$\hat{C}_j = \frac{1}{Q_j} \sum_{i \in \gamma(j)} A_{S,i} \exp\left(-\frac{L_{ij}}{v_{ij}\tau}\right) p_i, \qquad (1)$$

where $\hat{C}_j$, the modeled eDNA concentration of a given taxon at site $j$, results from the summation of eDNA production rates $p_i$ across all network nodes upstream of

$j$. These contributions are weighted by the relative source area $A_{S,i}$ and a first-order decay factor, where $L_{ij}$ is the along-stream length of the path joining $i$ to $j$, $v_{ij}$ the average water velocity along that path, and $\tau$ a characteristic decay time. Finally, $Q_j$ is a characteristic (mean during sampling days) value of water discharge across node $j$. Production rates are assumed to be proportional to the taxon's density and expressed via an exponential link

$$p_i = p_0 \exp\left(\boldsymbol{\beta}^T \boldsymbol{X}(i)\right), \qquad (2)$$

where $\boldsymbol{X}(i)$ is a vector of (normalized) environmental covariates evaluated at node $i$, $\boldsymbol{\beta}$ a vector of covariate effects and $p_0$ is a characteristic production rate[31]. Equation (2) represents a generalized linear model, a widely used class of species distribution models[70]. Note that Eq. (1) does not explicitly account for deposition of eDNA particles on the river bed. Decay and deposition are both relevant processes affecting eDNA removal, and thus detection, in stream water (see e.g., ref. [25]). We chose not to model these processes independently because it would be hardly possible to disentangle the roles of degradation of genetic material and that of gravity-induced deposition. Decay time $\tau$ is therefore to be seen as a characteristic time (linked to half-life $T_{1/2}$ by the relationship $T_{1/2} = \ln(2)\tau$) for the presence of eDNA in water, regardless of the source of its depletion.

We here further hypothesize that the expected number of reads for a given taxon at a given site is proportional to the taxon's eDNA concentration in the sample: $\hat{N}_j = k\hat{C}_j$. Equation (1) then becomes

$$\hat{N}_j = \frac{1}{Q_j} \sum_{i \in \gamma(j)} A_{S,i} \exp\left(-\frac{L_{ij}}{v_{ij}\tau}\right) p_0' \exp\left(\boldsymbol{\beta}^T \boldsymbol{X}(i)\right), \qquad (3)$$

where $p_0' = kp_0$. Finally, by denoting with $L_i$ and $w_i$ the length and width of reach $i$, respectively, it is $A_{S,i} = L_i w_i$, $v_{ij} = \sum_{k \in P_{ij}} L_k / \sum_{k \in P_{ij}} (L_k/v_k)$, where $P_{ij}$ is the set of nodes constituting the path joining $i$ to $j$. For the sake of clarity, all mathematical symbols used here and in the following are listed in Supplementary Table 2.

**Model calibration**. Measured numbers of reads at a given site are assumed to be distributed according to a geometric distribution with mean $\hat{N}_j$ obtained from Eq. (3). The geometric distribution is a special case of the negative binomial distribution, obtained by setting to one the parameter corresponding to the number of failures. Reasons for this choice are multifold: (i) it is a discrete distribution, as it is the case for read number data; (ii) it depends on a single parameter, thereby it reduces the complexity of the problem; (iii) it has null mode, which is in agreement with the data (in fact, the number of replicates with null read numbers ranged from 50 to 182 out of 183, depending on the genus); (iv) the distribution has fatter tails compared to the Poisson distribution, which is also a single-parameter, discrete distribution with null mode. Fat tails allow the interpretation of large observed numbers of reads.

Given these assumptions on the distribution of observed numbers of reads, the likelihood function reads

$$f\left(\boldsymbol{N}|\boldsymbol{\beta}, p_0', \tau\right) = \prod_{j \in S} \left[\prod_{o=1}^{3} \frac{1}{1+\hat{N}_j} \left(\frac{\hat{N}_j}{1+\hat{N}_j}\right)^{N_{jo}}\right], \qquad (4)$$

where $S$ is the set of sampling sites used to calibrate the model and $\boldsymbol{N}$ is a $|S|$-by-3 matrix whose entries are observed numbers of reads $N_{jo}$ at site $j$ and replicate $o$ ($|S|$ is the cardinality of $S$, equal to 61 in this case).

The posterior distribution of parameters $\boldsymbol{\beta}$, $p_0'$, $\tau$ was sampled by means of an Adaptive Metropolis algorithm[71]. The prior distribution adopted for $\boldsymbol{\beta}$ was a multivariate normal with null mean and covariance matrix equal to $9 * \boldsymbol{I}$, where $\boldsymbol{I}$ is the identity matrix of order equal to the number of components of $\boldsymbol{\beta}$. We then adopted a uniform prior distribution for $p_0'$ and a lognormal prior distribution for $\tau$ with mean equal to 2.55 h and standard deviation equal to 1.36 h, equivalent to a normal distribution for $\ln(\tau)$ (with $\tau$ expressed in seconds) with mean equal to 9 and standard deviation equal to 0.5; such prior distribution was derived from a previous study[31]. For each of the 50 model runs (corresponding to the 50 EPT genera detected in the eDNA samples), Markov chains were randomly initialized; the first 5000 elements of the chains built by the Adaptive Metropolis algorithm were then discarded (burn-in phase), while the following 10,000 were retained. With the above-mentioned settings, eDITH was used to generate spatial patterns of relative density for all of the 50 genera.

**Assessing detection probability and presence maps**. In order to enable the comparison among maps of relative density for the different genera, we resorted to the evaluation of detection probability. This was defined as the probability that, given the relative density of a genus at a given reach predicted by eDITH, an eDNA sample taken from that reach would yield a nonzero number of reads, if that reach were unconnected from the river network. From Eq. (3), the expected number of reads of a sample taken from an unconnected reach reads

$$\hat{N}_{U,i} = p_i' \frac{A_{S,i}}{q_i} \exp\left(-\frac{L_i}{v_i \tau}\right), \qquad (5)$$

where $q_i$ is the discharge directly contributing to reach $i$ (such that $Q_i = \sum_{k \in \gamma(i)} q_k$), $L_i$ and $v_i$ are length and water velocity relative to reach $i$, respectively. Thus,

according to the assumption of geometric distribution for observed read numbers, the probability of a non-zero read number for a sample with expected read number $\hat{N}_{U,i}$ is

$$P_{D,i} = \frac{\hat{N}_{U,i}}{1+\hat{N}_{U,i}}. \qquad (6)$$

Detection probabilities are evaluated by using median posterior values for parameters $\boldsymbol{\beta}$, $p_0'$, $\tau$. Finally, maps of detection probability are converted into presence maps by applying the threshold[36] $P_{D,i} \geq 2/3$. The summation of presence maps for all genera yields the genus richness map displayed in Fig. 5a.

A number of methods for threshold selection in species distribution models exist in the literature[72]; however, preliminary analyses (not reported) showed that methods such as the prevalence approach and the average probability/suitability approach (see ref. [72]) resulted in lower accuracy estimates (see below) as opposed to the fixed threshold approach. Moreover, the fixed threshold approach $P_{D,i} \geq 2/3$ was used by Mächler et al.[36]; since the eDITH model is to be seen as a way to translate "upstream-averaged" eDNA measurements into local equivalent eDNA measurements, it appears appropriate to keep the same criterion already used with the raw eDNA data.

For a given genus that was found both in the eDNA and kicknet samples, the accuracy of a presence map obtained by eDITH with respect to the kicknet data was determined as the fraction of true positives (i.e., sites where both model and kicknet assessed presence) and true negatives (i.e., sites where both model and kicknet assessed absence) over all (60) sites where kicknet was performed. On the contrary, false positives are sites where eDITH predicts presence while kicknet indicates absence; the vice versa holds for false negatives.

We underline that the eDITH model was run for all genera that were found in at least one replicate at one site. Such an inclusive choice was operated in a bid to maximize the available information and avoid false absences. However, the estimates of occurrence of taxa performed by our model remain conservative: indeed, for the 8 genera that were never found at any site with at least 2 out of 3 nonzero read numbers (see Supplementary Data 1), the fraction of river reaches where eDITH predicted presence was always lower than 3%.

Note that calculating the accuracy between model predictions and eDNA data with the method used for the model vs. kicknet comparison would be formally wrong: indeed, eDNA data are an aggregate measure of the upstream taxon distribution, while eDITH-based presence estimates refer to a local variable, because upstream contributions have been disentangled by the model. Conversely, an adequate measure of consistency between model predictions and eDNA data must compare quantities of the same type (i.e., both referred to the upstream catchment). To this end, we utilized the goodness-of-fit test detailed below.

**Goodness-of-fit test**. An ad hoc goodness-of-fit test was required owing to the use of a geometric distribution to represent the observed numbers of reads. The test relies on a bootstrapping technique derived from that proposed by Mi et al.[73], to which the reader is referred for details on the theoretical foundation of the method.

For a given site $j$, let $\hat{N}_j$ be the read number predicted by eDITH (see Eq. (3)) and $N_{jo}$, $o = 1,2,3$ the triplet of observed read numbers at that site. The sample standard deviation of the data with respect to the predicted mean reads

$$s_j = \sqrt{\sum_{o=1}^{3} \left(N_{jo} - \hat{N}_j\right)^2}, \qquad (7)$$

while Pearson's residuals are given by $r_{jo} = \left(N_{jo} - \hat{N}_j\right)/s_j$. Now, let $\tilde{N}_j \sim \text{Geom}\left(\hat{N}_j\right)$ be a random variable that is geometrically distributed with mean equal to $\hat{N}_j$. Let us generate a large ($h = 1, \ldots, 10^5$) number of triplets $\tilde{N}_{jho}$, $o = 1, 2, 3$ and compute their sample standard deviation $\tilde{s}_{jh}$ and residuals $\tilde{r}_{jho}$:

$$\tilde{s}_{jh} = \sqrt{\sum_{o=1}^{3} \left(\tilde{N}_{jho} - \hat{N}_j\right)^2}; \quad \tilde{r}_{jho} = \frac{\tilde{N}_{jho} - \hat{N}_j}{\tilde{s}_{jh}}. \qquad (8)$$

Let now $d_j$ and $\tilde{d}_{jh}$ be the sum of squared deviations of ordered residuals (of the data and of the sampled distributions, respectively) from the medians of their sampled distributions

$$d_j = \sum_{o=1}^{3} \left(r_{jo} - \tilde{r}_{jo}^{(50)}\right)^2; \quad \tilde{d}_{jh} = \sum_{o=1}^{3} \left(\tilde{r}_{jho} - \tilde{r}_{jo}^{(50)}\right)^2, \qquad (9)$$

where $\tilde{r}_{jo}^{(50)}$ is the median of the residuals of the $o$-th components of the generated triplets $\tilde{N}_{jho}$. The $p$-value can thus be computed as

$$p_j^{\text{GOF}} = \frac{1 + \sum_{h=1}^{10^5} \mathbf{1}_{\tilde{d}_{jh} \geq d_j}(h)}{1 + 10^5}, \qquad (10)$$

where $\mathbf{1}_{\tilde{d}_{jh} \geq d_j}(h)$ is an indicator function equal to one if $\tilde{d}_{jh} \geq d_j$ and null otherwise. We finally assumed that the null hypothesis $H_0$ that the triplet $N_{jo}$, $o = 1, 2, 3$ is

geometrically distributed with mean $\hat{N}_j$ (namely, that the model correctly reproduces the data at site $j$) cannot be rejected if $p_j^{GOF} > 0.05$.

**Cross-validation analysis of the effect of sample size**. In order to evaluate how model predictions vary as a function of data availability, we performed 9 additional simulations (termed AS), in which only subsets of the sampling sites were used to calibrate eDITH: 3 of them (simulation group termed AS1) were based on 49 out of 61 sites (~80%), 3 (AS2) were based on 37 sites (~60%) and 3 (AS3) were based on 25 sites (~40%). For each simulation, a quasi-random choice of the excluded subset of sites was operated. In particular, to ensure spatial coverage of the catchment, we first imposed that, for each AS group, the distribution of Strahler stream order values of the calibration subset was proportional to that of the complete set of sites (shown in Supplementary Fig. 2). The allocation of number of excluded sites per stream order value was determined via the D'Hondt method[74]. The excluded sites were then randomly sampled while respecting the constraint on stream order-based allocation. Note that, for a given thus-obtained calibration set, eDITH was run for all 50 genera.

For these additional simulations, calibration was performed as described with regards to the complete model (CM) (see "Model calibration" section), except that Markov Chains were here shorter (burn-in phase: 1000 elements; length of the retained chain: 5000 elements) to speed up the computational process.

Finally, performance indices were calculated in analogy to CM. Specifically, these were goodness of fit and accuracy. Goodness of fit is defined as the ability of the modeled spatial patterns of expected read numbers to reproduce the observed read numbers, and is evaluated as the fraction of sites (either all sampling sites, calibration or validation subsets) where the null hypothesis that observed read numbers come from the hypothesized distribution cannot be rejected (see "Goodness-of-fit test" section). Accuracy is defined as the agreement between model-based genus presence predictions and kicknet observations, and is evaluated as the fraction of sites marked as true positives or true negatives (see "Assessing detection probability and presence maps" section). We then calculated loss of goodness of fit as the difference between the goodness of fit of the AS simulations and that of CM. Loss of accuracy is analogously calculated.

**Reporting summary**. Further information on research design is available in the Nature Research Reporting Summary linked to this article.

## Data availability

Sequence data that support the findings of this study have been deposited in European Nucleotide Archive with the study accession numbers (secondary accession number) PRJEB31920 (ERP114535) and PRJEB33506 (ERP116301). Hydrological and landscape data that support the findings of this study are available in Zenodo with the identifier https://doi.org/10.5281/zenodo.3903330. Source data are provided with this paper.

## Code availability

MATLAB scripts reproducing the results of this manuscript are available at https://doi.org/10.5281/zenodo.3903330. Source data are provided with this paper.

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

## Acknowledgements

The data analyzed in this paper were generated in collaboration with the Genetic Diversity Centre (GDC), ETH Zurich. We thank Roman Alther, Simon Flückiger, Emanuel A. Fronhofer, Isabelle Gounand, Sereina Gut, Eric Harvey, Samuel Hürlemann, and Chelsea J. Little for help in the field and lab work. We thank Jean-Claude Walser for help with the sequencing analyses. Funding is from the Swiss National Science Foundation Grants No PP00P3_179089 and 31003A_173074, and the University of Zurich Research Priority Program "URPP Global Change and Biodiversity" (to F.A.).

## Author contributions

L.C. developed the model and analyzed the data. E.M., R.W., and F.A. collected the data. L.C. and F.A. led the writing of the manuscript, with critical contributions from all co-authors. All authors approved the final version of the manuscript.

## Competing interests

The authors declare no competing interests.
