## [Peer Review File · Nature Communications]

Reviewers' Comments:

Reviewer #1:

Remarks to the Author:

Carraro et al. use a large dataset on EPT taxa with parallel data from 60 sites using kicknet sampling and eDNA metabarcoding of water sampling across a large river basin. They use the eDNA dataset for a modelling approach integrating transport and hydrology to derive estimates of richness in maps at a much finer level of spatial resolution than just the point estimates.

The paper provides a very appealing approach and the results are convincing as are the impressive extent of the dataset. It is thus a novel and important contribution to the eDNA literature, which truly has implications for the fields of freshwater ecology, biodiversity and conservation.

I think the paper is worth publishing (perhaps also in a high-impact journal), but it should be noted that the dataset has been published before with a slightly different focus in (Mächler et al. 2019. Environmental DNA). I have a few suggestions on how to improve it /make it more relevant to a broader readership.

Firstly, I am missing some discussion on the individual taxa found in the study and the covariates. The paper gives important information on which covariate have positive or negative effect on each EPT genus, but does this make biological sense in terms of the ecology and life history of the individual genus/species? Some examples of genera with distinct ecology could be very beneficial. For example, some species are associated with forest, some with higher elevation etc.

Also, a more elaborate discussion of the fact that the model underestimates richness in higher order streams would be very useful. What does this mean for applying the approach in management? Should the point estimates for eDNA just be used instead for higher stream orders, since this estimate seems to match the kicknet data better? (Figure 4). Or can any addition to the model potentially cope with this discrepancy?

A general discussion on where this could be implemented in management could also be beneficial, including a discussion on the potential of the model to be generalized to other taxonomic groups such as fx. freshwater fish or molluscs.

Minor suggestions:

I suggest to use "eDNA" in the title, so it is more obvious to the reader what the study is about.

L. 70 "50 EPT genera, present in at least one replicate at one site."

One replicate is very little, and is more prone to accepting sequencing errors as true presence of taxa. This is also in conflict with the text later on in figure 3: "Genus richness from eDNA data; here presence of a genus at a site is attributed if at least 2 out of 3 replicates have nonzero read numbers" Are different criteria used for detection of the 50 taxa and data for the Genus richness maps? They should be consistent.

L. 105: "...and estimating characteristic decay times of eDNA (mean across all genera: 2.6 h; the distribution of values is shown in Extended Data Fig. 2)."

2.6 h seems very short for degradation according to the eDNA literature, where estimates of days are typical, and so the results are in contrast to the literature. What does "decay time" mean in this context exactly? Is it the time from eDNA release until it can no longer be detected?

Please elaborate and discuss if this result could potentially be underestimating the decay time.

L. 188. Explain abbreviation LUT (Luteren)

L. 205: "We here further hypothesize that the expected number of reads for a given taxon at a given site is proportional to the taxon's eDNA concentration in the sample".

I would also expect this in general, but primer affinity bias can affect it. Please discuss the primers used in this study in terms of tests for affinity across EPT taxa. The primers used are COI, which are especially prone to differences in affinity (See fx. Deagle et al. 2014. Biol. Letters). Please refer to literature on this topic.

Fig. 3. Explain the black circles in 3a?

Reviewer #2:

Remarks to the Author:

This study used the collected eDNA and kicknet data at Thur catchment/basin to build the species distribution model (SDM) for 50 EPT genera. The efforts of this study (both the field and computation) are impressive. The aim of building the SDM by associating the hydrology, landscape factors, and the eDNA production is applaudable. However, the clarity of the study description might have undermined the value of this paper. I value this study and hope the questions and concerns below can be addressed and the clarity of paper can be improved.

Here are a few main concern/questions to improve the clarity:

#The site description and data collection-----

The number of sampling locations and their locations choices are critically important to building SDM in terms of their representative, especially when doing this spatially explicit species distribution model.

The total length of the stream and how the survey sites distributed were not clearly defined at first.

Line 43: These eDNA samples have been taken by the stream bank. Taking "at a river's cross-section (Line 43)" seems misleading.

Line 61: Taxa's "relative abundance (Line 61)" typically used to inference the taxa abundance within one site sample. Using "relative abundance" to describe modeling results (either for one species or one out of 50 EPT genera) seems misleading.

Line 147: "250 ml of river water were filtered three times on a separated filter..." Does this mean the same water filtered 3 times or 3 replicates?

Noticing one of the coauthors had collected and contributed these data (i.e., eDNA, Kicknet samples from Mächler et al. 2018), the coauthor might have extensive knowledge to help address some of the confusion.

Hydrological Data and Principle

Authors claim this is a "hydrology-based model," while the actual incorporation of flow information seems to only based on "median discharge or median water velocity."

Line 21: It is not clear what the authors mean by "hydrological first principles." There is no such term described in general hydrological science or in the referred citation 13 (Carraro et al. 2018).

Line 168-174: the power law regressions (of Q, D, w, v) were derived based on 4 stations. This is very questionable! Literally, if you plot them out, using four points try to find the power law regression between Q vs A, D vs A and w vs A. I very much like to see author provide the plots and statistics on how well did they fit the power law regression. Authors' claim on the uniqueness of this model relies on the hydrological model/relationship.

#eDNA reads vs taxa concentration/abundance/presence threshold

Line 80-82: The assumption is the read number is proportional to the eDNA concentration/prediction. Knowing little the relationship between read number and eDNA concentration, I very much like to believe this assumption is reasonable and true. I would appreciate author to provide more

justification/citation/literature review.

Line 61-62: local abundance. eDNA is highly criticized for use to represent "abundance." It is recommended to use to indicate species presence/absence. Similar as above, please provide justification for using eDNA, and the model built with eDNA for taxa abundance.

#Landscape variables/covariates

The landscape covariates can be local or upstream scale, which make a lot of sense (as eDNA is likely the accumulated results from upstream and local contribution).

Line 184-185: The additional "geographical" covariates was not justified. Not sure why this constraint was needed, especially eDNA is results is accumulated from upstream and is accounted in the model.

Model results/interpretation

Line 252: using an arbitrary 2/3 threshold might not be the best strategy in SDM. Please consider Liu, C., Berry, P.M., Dawson, T.P. and Pearson, R.G. (2005), Selecting thresholds of occurrence in the prediction of species distributions. *Ecography*, 28: 385-393. doi:10.1111/j.0906-7590.2005.03957.x

Line 258-259: Why do you treat kicknet results as the true presence, and compare modeling result against kicknet? The model was built with eDNA. Are they always consistent with the eDNA sample at the given sites?

Detailed comments attached.

Hope this helps. Thanks for the opportunity to review the paper and provide the feedback/comments.

Reviewer #3:

Remarks to the Author:

My overall comment is that the work will be of broad interest, but that the presentation is so brief as to be incomplete. This makes the work very difficult to evaluate. Simply put, the work seems very good and topic very interesting, but the article is too short and lacks necessary details. If a word limit is imposed by the journal and can not be modified, then I suggest another outlet would be more suitable. Most of comments, as you will below, have do do with there not being enough information, or space to even suggest some alternative explanations, hypotheses, or considerations.

Line 51: It is not clear the role of hydrology in the paper and in the model. Deposition is a very important factor in particle (i.e., non-solute) transport, and it is not clear whether this is dealt with in your model here, or whether previous work has investigated the role of deposition and can quantify it or rule it out as negligible. This section needs some more detail aside from a few (limited) factors listed in parentheses.

Line 54 (also line 75) The status of the model is unclear throughout. Is it an existing model or is being newly described here? The text gives the impression that it exists and is fully described in ref. 25; however, the methods section spends a great deal of time describing it. I was not sure whether I am peer-reviewing the model itself or whether it has been reviewed and I am reviewing its application to eDNA. I think a few lines of explanation of the model and its history/previous applications would help here.

Line 60: How were these maps produced?

Line 61: awkward wording: perhaps "of each taxon's" of "of the presence of each taxon"

Line 63: These samples are not described. How many sites? At first it looks like 61 sites (line 65) but later only 60 sites (line 257). Are the same kick-net samples used for both purposes? Again, a bit more text would help here. The citation is for the methodology, but as used here looks like these data are published. Please clarify that these are also newly collected data.

Line 65: Change "Threefold replicated" to "Three eDNA and three invertebrate..." These are, at best, technical replicates.

Line 67: I find this too brief, even in the method section there are too many details missing.

Line 72: I would like to see some evaluation of whether this is a suitable read depth. What does "pooled over genera and replicates" mean? How were data handled i.e. if a genus was found with 2 reads in only one "replicate", 10 reads in two replicates, etc(?)

Line 77: What is the source of the species distribution model? Is it constructed from the kicknet sample results?

Line 79: Please give some detail regarding the "uncertainties". I did not find it obvious which part of the methods what this was referring to. There are many that are never dealt with (PCR bias, contamination, ...). At a minimum these could be mentioned in a discussion.

Line 104: Please define "biodiversity in general" more specifically - this refers to the fig.2 covariate analysis which is not mentioned anywhere except in the methods.

Line 113: Can the authors rule out DNA from upstream sources, or cross-contamination of samples as alternative explanations here(?) Alternative explanations are rarely offered in the text.

Line 115: Please give criteria for a "match" and add this to fig legend.

Line 116: Is there a known relationship? If so please elaborate.

Line 119-124: Are there alternative explanations? This section is not easy to understand without any explanation of the model itself, which is mostly in "methods". Why and how are downstream sites more prone to "error" - does error mean false positives, negatives, both?

Line 136: What are these first principles? At a minimum, a short description of the model and how it works with eDNA data (and why kick-net data would not be suitable). Why not consider the fact that kick-net samples might also have individuals that have drifted from upstream, and may not consist of the local community. I am missing some ecological considerations of the approach and its potential limitations.

Line 148: Please give details of the filters and extraction methods.

Line 152: If this pipeline is not published or described, then it is impossible to evaluate the approach. Please provide at least minimal information on the analysis. I would be interested in how the multiple samples ("replicates") were handled - did an OTU (or ASV?) need to be in at least two to be considered present? etc.

Line 168: What is the time interval for which the median is calculated? Annual? Are four stations enough to adequately characterize the entire catchment? Some discussion of potential limitations are needed here and elsewhere.

Line 176: Seems to only be in the legend of Fig. 2 and does not add anything to the study as presented. My apologies if i missed it in the text other than line 104-5.

Line 201: How are these densities estimated for each taxon at each site? From kicknet samples? Are these quantitative enough? Are there other data available?

Line 123: This is an interesting approach (geometric distribution). Has it been used elsewhere? Was there any test as to whether it provided a better empirical fit than other distributions? Does the bioinformatic analysis pipeline need to consider this?

Line 265: There is a lot of important on the read numbers - the paper would benefit from some discussion as to how this might be problematic. See my comment re source of abundance data (above, comment re line 201).

Line 314: How old?

Line 335: I could not see the cross in my copy.

Line 338: Fig. 2a please use a more obvious y-axis title(?) At a minimum, state in the legend what H0 is. I would also try to add the genus names, to have a bit of ecology in the figure. Were there any patterns e.g. were the less accurate taxa always E, P, or T? Perhaps members of a family? As presented, the fig is not stand-alone. Fig. 2b,c is not helpful unless covariates are identified. I propose removing one panel and writing out the covariates. The panels are not adequately described in the text, and could even be removed.

NCOMMS-19-40091-T - Upscaling spatial patterns of biodiversity in freshwater ecosystems

by Luca Carraro, Elvira Mächler, Remo Wüthrich, Florian Altermatt

Response to Reviewers

We have addressed all the Reviewers' concerns with detailed replies (see below) and corresponding amendments in the manuscript text. We specifically thank the Reviewers for their comments, which allowed us to enhance the clarity of our approach and results.

We hereafter outline the major changes performed:

- The title was changed to *Environmental DNA allows upscaling spatial patterns of biodiversity in freshwater ecosystems*, following a suggestion from Reviewer #1.
- Thanks to a comment from Reviewer #2, we re-performed the derivation of hydrological variables with slightly changed initial assumptions. As a result, we re-conducted all subsequent analyses on taxon distributions and biodiversity assessment. Final results are qualitatively and quantitatively very consistent and have only very minimally changed with respect to the former manuscript version.
- We created Results and Discussion sections in the manuscript, where we addressed the relevant points raised by the Reviewers. In particular, we added a paragraph in the Results section where we compared the distribution of some relevant genera as predicted by the model with the relative ecological and faunistic knowledge. Moreover, in the Discussion, we highlighted advantages, limitations and perspectives of our study.
- As asked by all Reviewers, the Methods section was substantially expanded to include extensive details on field data collection, eDNA processing and sequencing, and the hydrological assumptions used.

All major changes and new text in the manuscript are highlighted in yellow.

Detailed replies follow below. Reviewers' comments are reported in *italicized blue type*. Our replies are in roman type. Excerpts from the amended manuscript are in *italicized type*. The acronym **RX** identifies our replies throughout this document, to allow cross-referencing.

Reviewer #1 (Remarks to the Author):

Carraro et al. use a large dataset on EPT taxa with parallel data from 60 sites using kicknet sampling and eDNA metabarcoding of water sampling across a large river basin. They use the eDNA dataset for a modelling approach integrating transport and hydrology to derive estimates of richness in maps at a much finer level of spatial resolution than just the point estimates.

The paper provides a very appealing approach and the results are convincing as are the impressive extent of the dataset. It is thus a novel and important contribution to the eDNA literature, which truly has implications for the fields of freshwater ecology, biodiversity and conservation.

I think the paper is worth publishing (perhaps also in a high-impact journal), but it should be noted that the dataset has been published before with a slightly different focus in (Mächler et al. 2019. Environmental DNA). I have a few suggestions on how to improve it /make it more relevant to a broader readership.

Reply 1 (R1): We wish to thank this Reviewer for the positive and constructive assessment of our work. We have integrated all suggestions. It is correct (as also prominently stated in the manuscript) that the technical aspects of the eDNA analysis have been published before, as this was a necessary technical basis for this manuscript. However, none of the approaches and results on upscaling biodiversity across whole communities of organisms in complete riverine networks have been published, nor are they under consideration elsewhere. Thus, the manuscript—especially with its relevancy for freshwater biology, biodiversity and conservation research—is completely novel.

Firstly, I am missing some discussion on the individual taxa found in the study and the covariates. The paper gives important information on which covariate have positive or negative effect on each EPT genus, but does this make biological sense in terms of the ecology and life history of the individual genus/species? Some examples of genera with distinct ecology could be very beneficial. For example, some species are associated with forest, some with higher elevation etc.

R2: This is indeed a good suggestion. We expanded the text and discussed the biological context of the species found in more detail. We have now added a paragraph in the results section where we discuss observed relationships of ecology and life history and the spatial distribution of representative taxa, and the associated effect of covariates (LL. 145-168):

“Remarkably, a fair agreement can be observed between the eDITH-based predictions of spatial distribution of taxa (Fig. 4), the related predicted role of covariates (Fig. 2a) and the ecological and faunistic knowledge on taxa [CSCF, 2020]. We here discuss the results obtained for three representative taxa. Only two species belonging to the mayfly genus Habroleptoides (H. confusa, H. auberti) are known to occur in Switzerland and in the Thur catchment. Habroleptoides auberti is reported in the upper Thur and Necker basins, which largely reproduces the patterns displayed in the left column of Fig. 4. Habroleptoides confusa is reported to occur in waters of neutral pH, which may explain the negative relation of swamps (covariate L-SW) and peat (G-PE) found by the model (see line 31 of Fig. 2a). Both species are sensitive to pesticides, which justifies the predicted negative role of urban areas (L-UR), drainage area (L-DA) and orchards (L-OR). As for the stonefly Protonemura, three species (P. brevistyla, P. auberti and P. nitida) are expected to be present in larval stages in the Thur catchment in late June (when the sampling was performed), while the emergence of P. lateralis is likely to occur earlier in the season. All three species are commonly found in spring brooks, epirhithral and metarhithral (i.e. upper and middle upland) streams, but also in unpolluted waterbodies. This is in agreement with the predicted distribution maps shown in the central column of Fig. 4 and with the positive role found for scree (G-SC, see line 50 of Fig. 2a), local elevation (M-LE) and the upper geographical clusters of the Thur (TH6, TH7, TH8—compare with Fig. 1b). Finally, the stonefly Athripsodes is mostly represented by the species A. albifrons in the study catchment. This eurythermal species is typically found in lower rhithral and epipotomal reaches, characterized by midsize to large river widths and warm summer temperatures. The predicted distribution maps for Athripsodes (right column of Fig. 4) and the positive role found for geographical clusters TH1, TH2, TH4 and NE2 (see line 14 of Fig. 2a and Fig. 1b) match such empirical observations, although model results predict absence of Athripsodes in clusters TH3 and NE1, which are also plausible habitats for this taxon.”

Also, a more elaborate discussion of the fact that the model underestimates richness in higher order streams would be very useful. What does this mean for applying the approach in management? Should the point estimates for eDNA just be used instead for higher stream orders, since this estimate seems to match the kicknet data better? (Figure 4). Or can any addition to the model potentially cope with this discrepancy?

R3: This is an interesting comment. We added a paragraph on this aspect in the discussion section. It is important to note that, while eDNA data shows genus richness closer to the kicknet one in high order streams as compared to the model, the accuracy of eDNA alone in predicting presence/absence of a specific genus would likely be worse than that of the model. Indeed, the more downstream a sampling site is located, the more the corresponding eDNA sample integrates inputs from different sources and along-stream paths, and is therefore less representative of the local community. We hence argue that more eDNA data collected downstream (if possible, downstream of the outlet, i.e. the most downstream sampling site(s) should not belong to the region where the model is used for assessment) would improve model estimates, which also means, from a management perspective, that downstream sites are most valuable for estimates on spatially integrated diversity estimates. We clarified this concept in the revised version of the manuscript (LL. 217-230):

“In our case study, biodiversity predictions produced by eDITH proved to be rather robust to the choice of sampling sites, as highlighted by our cross-validation analysis (see Fig. 7c,d and Table 1). However, the subsamples of sites used in such investigation were chosen such that the proportions of sites from the upstream and downstream portions of the catchment (reflected by the stream order values of the corresponding reaches) were not altered with respect to the original set of sites. We also observed that including information from downstream sampling sites in the eDITH model is instrumental in evaluating taxon abundance in the upstream reaches (due to eDNA transport), but generally leads to poor prediction of taxon abundance at local, downstream communities (see Fig. 6). We therefore suggest using our model to assess biodiversity in the portion of the catchment located some kilometers upstream of the most downstream eDNA sampling site. Measurements taken at this site are still needed for estimation of the upstream pattern of taxon distribution, but will not be informative enough to disentangle the contribution to the eDNA signal from the nearest sources. Moreover, further studies should investigate how the positioning of eDNA sampling sites within a catchment influences the prediction power of the eDITH model.

A general discussion on where this could be implemented in management could also be beneficial, including a discussion on the potential of the model to be generalized to other taxonomic groups such as fx. freshwater fish or molluscs.

R4: We wish to thank this Reviewer for this insightful suggestion. We extended both on the overall potential of the method as well as its generalization to other taxonomic groups. We specifically added the following paragraph to the discussion (LL. 193-204):

“Our framework opens new avenues for freshwater ecosystem management, as it may allow non-invasive and efficient localization of e.g. endangered or invasive taxa, pathogens, as well as biodiversity assessments over taxonomic groups wider than those here used as case study. Indeed, applying eDITH to motile organisms (such as freshwater fish, crustaceans or mollusks) is possible because the time scale of hydrological transport in rivers (say, considering a reference water velocity of 1 m/s) is much faster than that of mobility of most organisms, at least over wide distances. In principle, application to terrestrial taxa inhabiting areas with high drainage density of the stream network is also possible. In this case, the source area A_S (see Eq. (1)) should be defined as the subcatchment area, rather than the riverbed area as in the present case. However, further caution in the modelling approach would be required, as only a fraction (possibly dependent on the distance from the drainage network) of the eDNA shed by such organisms will be transported downstream by the streamflow.”

Minor suggestions:

I suggest to use “eDNA” in the title, so it is more obvious to the reader what the study is about.

R5: This is a good suggestion. The title now reads: “*Environmental DNA allows upscaling spatial patterns of biodiversity in freshwater ecosystems*”

L. 70 “50 EPT genera, present in at least one replicate at one site.” One replicate is very little, and is more prone to accepting sequencing errors as true presence of taxa. This is also in conflict with the text later on in figure 3: “Genus richness from eDNA data; here presence of a genus at a site is attributed if at least 2 out of 3 replicates have nonzero read numbers” Are different criteria used for detection of the 50 taxa and data for the Genus richness maps? They should be consistent.

R6: This may be a misunderstanding, which we have now clarified. We were inclusive in the taxa considered to avoid false absences, but low recovery rates were taken into consideration by the model and resulted in conservative estimates. Specifically, the EPT genera analyzed by the eDITH model were those that were detected in at least 1 out of 3 replicates in at least one site. Such choice was done in the spirit of maximizing the information contained in the data. If, however, a genus G were only detected in a single replicate at a single site and were undetected at all other sites (which could represent, as this Reviewer correctly states, a false positive), subsequent model estimates of relative density for G would be extremely low, which would lead to estimation of detection probability for G close to 0. This is actually the case for the genus *Chaetopteryx*, which is estimated as absent from the whole river network based on the fixed-threshold approach. On the other hand, if read numbers for a genus G_2 across several sites are of the form $[x, 0, 0]$ (with $x > 0$), in this case the model is more likely to identify a possible source habitat for G_2 . For example, this is the case for the genus *Ephemerella*, which was found in 5 sites via eDNA, and in all cases only 1 replicate out of 3 yielded a nonzero read number. The eDITH model predicted presence of *Ephemerella* in 16 (out of 760) reaches of the catchment. We have now added the following paragraph in the Methods section (LL. 467-472):

“We underline that the eDITH model was run for all genera that were found in at least one replicate at one site. Such an inclusive choice was operated in a bid to maximize the available information and avoid false absences. However, the estimates of occurrence of taxa performed by our model remain conservative: indeed, for the 8 genera that were never found at any site with at least 2 out of 3 nonzero read numbers (see Supplementary Table 1), the fraction of river reaches where eDITH predicted presence was always lower than 3%.”

L. 105: “...and estimating characteristic decay times of eDNA (mean across all genera: 2.6 h; the distribution of values is shown in Extended Data Fig. 2).” 2.6 h seems very short for degradation according to the eDNA literature, where estimates of days are typical, and so the results are in contrast to the literature. What does “decay time” mean in this context exactly? Is it the time from eDNA release until it can no longer be detected? Please elaborate and discuss if this result could potentially be underestimating the decay time.

R7: Decay time τ is here defined as the parameter of a first-order decay function: $C(t) = C_0 \exp(-t/\tau)$, which implies that, when $t = \tau$, the concentration in the sample is equal to $e^{-1} C_0 \cong 0.367 C_0$, where C_0 is the concentration at $t = 0$. Decay time is linked to the half-life $T_{1/2}$ by the relation $T_{1/2} = \tau \ln(2)$. Note that the value of mean (across genera) decay time for the new analysis is $\tau = 1.5$ h; a lower value with respect to the previous one is due to the use of increased values of water velocity (see replies **R16-R17** to Reviewer #2 with respect to the scaling of hydrological variables). With $\tau = 1.5$ h, the concentration after 6 h from release is about 2% of the original concentration, and presumably still detectable. Higher values of τ , as the one estimated for *Caenis*, $\tau = 6.9$ h, are such that, after 24 h, C is higher than 3% of C_0 . Importantly, as pointed out by Reviewer #3 (see **R40**), we do not explicitly account for deposition of eDNA. Therefore, estimates of τ account for all possible sources of eDNA removal from stream water,

which explains why our estimates for τ are somehow lower than those reported in lab studies where pure dynamics of eDNA degradation are assessed. We have now expanded on the concept of decay time in the manuscript (see reply **R40**).

L. 188. Explain abbreviation LUT (Luteren)

R8: Done.

L. 205: "We here further hypothesize that the expected number of reads for a given taxon at a given site is proportional to the taxon's eDNA concentration in the sample". I would also expect this in general, but primer affinity bias can affect it. Please discuss the primers used in this study in terms of tests for affinity across EPT taxa. The primers used are COI, which are especially prone to differences in affinity (See fx. Deagle et al. 2014. Biol. Letters). Please refer to literature on this topic.

R9: We now expanded the discussion about the primers used and the implications from a modelling perspective. Please refer to our reply **R18** to Reviewer #2 on the same subject.

Fig. 3. Explain the black circles in 3a?

R10: We modified the legend of Fig. 3 (now Fig. 5) as requested.

Again, we wish to thank this Reviewer for her/his constructive suggestions.

Reviewer #2 (Remarks to the Author):

This study used the collected eDNA and kicknet data at Thur catchment/basin to build the species distribution model (SDM) for 50 EPT genera. The efforts of this study (both the field and computation) are impressive. The aim of building the SDM by associating the hydrology, landscape factors, and the eDNA production is applaudable. However, the clarity of the study description might have undermined the value of this paper. I value this study and hope the questions and concerns below can be addressed and the clarity of paper can be improved. Here are a few main concern/questions to improve the clarity:

R11: We thank this Reviewer for her/his positive and constructive feedback. We agree with all her/his suggestions, and have especially improved the clarity of the manuscript, especially including more details on site selection, data processing and modeling approaches.

#The site description and data collection-----

The number of sampling locations and their locations choices are critically important to building SDM in terms of their representative, especially when doing this spatially explicit species distribution model. The total length of the stream and how the survey sites distributed were not clearly defined at first.

R12: We have revisited the method section and ensured that all the relevant information is provided. Please see LL. 327-328: "...the channelized (i.e. perennial, see O'Callaghan and Mark [1984]) portion of the drainage network, whose total length equaled 751 km". We added the following sentence on site selection in the Methods section (LL. 275-277): "Sampling sites were chosen in order to proportionally represent all stream orders in the river network (Supplementary Figure 1) and to span the complete geographical extent of the catchment." Supplementary Figure 1 (reported below as Fig. R1) has been added to the supplement.

Figure R1. Distribution of stream order values across the 61 reaches where eDNA sampling sites were located (left panel) and across the 760 reaches constituting the Thur river network (right panel). The site where eDNA sampling was performed but kicknet was not has stream order value equal to 5.

We also added a paragraph on the robustness of model outcomes with respect to the choice of sampling sites (see reply **R3** to Reviewer #1).

Line 43: These eDNA samples have been taken by the stream bank. Taking "at a river's cross-section (Line 43)" seems misleading.

R13: We apologize for this possible misunderstanding. The paragraph mentioned referred to a general consideration on the phenomenon of downstream transport of eDNA within river networks, and not

our sampling from the stream bank. For the sake of clarity, we rephrased it using “site” instead of “cross-section”.

Line 61: Taxa’s “relative abundance (Line 61)” typically used to inference the taxa abundance within one site sample. Using “relative abundance” to describe modeling results (either for one species or one out of 50 EPT genera) seems misleading.

R14: We are not completely sure if we understand this comment correctly. We think this comment refers to an additional comment added in the annotated PDF by the reviewer. The corresponding comment on the PDF reads “*eDNA is mostly used to detect present/absence. The use of abundance is not recommended*”. Our reply will refer to this latter comment, but please let us know if we have misunderstood the comment.

The use of abundance vs. presence/absence data from eDNA studies is indeed highly debated. Recent work (e.g., Buchner et al. [2019]) showed that, for bulk-sample metabarcoding, conclusions on river ecological status classification do not differ substantially between presence/absence vs. abundance based estimates, and such observation is also directly applicable to eDNA data. Thus, we agree that, for the direct assessment of ecological status from eDNA data, the use of a presence/absence criterion would be justifiable.

However, we need to take abundances into account for our model, even though we do not claim nor directly infer the match of the eDNA abundance vs. actual organismal abundance. Please note that we compare the model and the eDNA-based data on a presence-absence base, even though the model has been run taking eDNA abundance into account. In fact, the principle of our model is that, in order to be able to disentangle the upstream contributions to the eDNA signal at a given river site, it is necessary to think in terms of quantities of eDNA shed in the different sources. Based on well-established assumptions, we then formulated the model in the present form. Our (arguably idealized) approach assumes that locally produced eDNA is found (on average) at a higher number than distantly produced. Based on basic principles of chemistry, such a distance-decay is the most realistic assumption to make. We now state more clearly that we only make conclusions on the occurrence of organisms at the level of presence-absence, which is a conservative, robust and generally supported level to look at, also with eDNA data.

We acknowledge that the processes of shedding, transportation and decay of eDNA are affected by many uncertainties (not to mention the uncertainties related to eDNA extraction and sequencing), which could not all be addressed by our assumptions. However, using conservative assumptions on a distance-decay makes our approach useful and robust, as it allows providing reliable estimates of taxa presence/absence (as well as reasoned guesses about the taxa’s relative abundance distribution), which would not be possible without a quantitative/phenomenological interpretation of the eDNA data.

The following paragraph has been added to the Discussion section (LL. 255-262): “*Finally, it is worthwhile to note that a quantitative interpretation of eDNA data in rivers is crucial even if the ultimate goal is to make biodiversity predictions based on estimates of presence or absence of the investigated taxa. Indeed, in order to understand eDNA advection and decay dynamics across a river network, it is essential to adopt a mechanistic approach such as eDITH, which frames the problem of eDNA transport in terms of quantities of eDNA shed in the different upstream sources. This enabled us to derive predictions of spatial patterns of relative taxon density (Fig. 4); however, as a conservative assumption, both the ground-truthing of model predictions against the kicknet data and biodiversity estimates were operated on a presence/absence basis.*”

Line 147: “250 ml of river water were filtered three times on a separated filter...” Does this mean the same water filtered 3 times or 3 replicates? Noticing one of the coauthors had collected and contributed these data (i.e., eDNA,

Kicknet samples from Mächler et al. 2018), the coauthor might have extensive knowledge to help address some of the confusion.

R15: We apologize for not having been specific enough on the field sampling. We have now rephrased this paragraph. We used three independent water samples, that is, we independently filtered 3 times 250 mL, and each of these 250 mL was filtered on a separate filter. See also reply **R45** to Reviewer #3.

Hydrological Data and Principle

Authors claim this is a “hydrology-based model,” while the actual incorporation of flow information seems to only be based on “median discharge or median water velocity.”

Line 21: It is not clear what the authors mean by “hydrological first principles.” There is no such term described in general hydrological science or in the referred citation 13 (Carraro et al. 2018).

R16: We have revisited this section and clarified the terminology. Our model is indeed a hydrology-based model, and we apologize if we have used jargon from hydrology that was not describing this in detail. We have now better described that the major hydrological aspects used in our model.

As widely known in hydrology, river width w , river depth D and water velocity v all scale (both within a single cross-section and in the downstream direction) as a power-law of water discharge Q [Leopold and Maddock, 1953, Leopold et al., 1964]. Moreover, water discharge scales linearly across a catchment with drainage area [Rodriguez-Iturbe and Rinaldo 2001]. Strictly speaking, the relationship $Q \sim A$ holds for mean annual values of Q , but can be reasonably extended to values of Q averaged over shorter time windows (say, one day), provided that the time scale of flow propagation is much shorter than one day, and that rainfall (and the resulting runoff generation) can be considered spatially homogeneous if aggregated at a daily scale [Carraro et al., 2020].

Importantly, these hydrological scaling relationships allow deriving universally valid links that are indeed based on averages and do not depend on hydrological variables measured at high spatial and temporal resolution. The relationships considered here are valid for homogeneous climatic and hydrological conditions. We now specify this aspect in the methods section, and highlight that eDNA samples were not affected by singular rain events (LL. 274-275). The approach used is instrumental in deriving values of the relevant hydrologic variables (Q , w , D , v) at any point in the catchment, without which it could not be possible to write Eq. (1) of the model.

The Methods paragraph has been modified as follows (LL. 340-363), including excerpts from the above reply:

“In order to assess values of hydrological variables for all reaches, we made use of power-law scaling relationships, a well-established and universally applicable concept in hydrology [Leopold and Maddock, 1953, Leopold et al., 1964, Rodriguez-Iturbe and Rinaldo, 2001]. In particular, river width w , river depth D and water velocity v are known to scale (both within a single cross-section and in the downstream direction) as a power-law of water discharge Q [Leopold and Maddock, 1953; Leopold et al., 1964]; along the flow direction, the relationships $w \sim Q^{0.5}$, $D \sim Q^{0.4}$, $w \sim Q^{0.1}$ are valid over wide ranges of natural streams [Leopold and Maddock, 1953]. Moreover, water discharge scales linearly across a catchment with drainage area A [Rodriguez-Iturbe and Rinaldo, 2001]. Strictly speaking, the relationship $Q \sim A$ holds for mean annual values of Q ; however, it can be reasonably extended to values of Q averaged over shorter time windows (say, at least one day), provided that the time scale of flow propagation is much shorter than one day, and that rainfall (and the resulting runoff generation) can be considered spatially homogeneous if aggregated at a daily scale. Both assumptions are reasonable for catchments up to that 10^3 km^2 [Carraro et al., 2020], as the one here studied. Mean water discharges during the sampling days and stage-discharge relationships

were available at four stations operated by the Swiss Federal Office for the Environment (FOEN) (see Fig. 1). River widths at these locations were estimated via aerial images. Power law relationships with drainage area for discharge and river width were then fitted on the four stations, yielding $Q = 0.072A^{1.056}$ and $w = 1.586A^{0.526}$, where A is in km^2 , Q in m^3s^{-1} , and w in m . As for river depth, we discarded the station with lowest drainage area because we observed that the values of depth measured therein were highly overestimated with respect to the expected values, based on the other stations and the scaling exponent 0.4 [Leopold and Maddock, 1953]. Hence, we limited the fit to the three other stations and obtained $D = 0.073A^{0.463}$, where D is in m . By assuming rectangular river's cross-sections, we finally derived a power-law relationship linking water velocity v to drainage area: $v = Q / (Dw) = 0.623A^{0.067}$, where v is in ms^{-1} . Notably, all scaling exponents obtained were very close to the literature values [Leopold and Maddock, 1953; Rodriguez-Iturbe and Rinaldo, 2001]. Details on the fit of these hydrological relationships are reported in Supplementary Figure 2."

Line 168-174: the power law regressions (of Q , D , w , v) were derived based on 4 stations. This is very questionable! Literally, if you plot them out, using four points try to find the power law regression between Q vs A , D vs A and w vs A . I very much like to see author provide the plots and statistics on how well did they fit the power law regression. Authors' claim on the uniqueness of this model relies on the hydrological model/relationship.

R17: We apologize again for not having been clear enough. These linear relationships are well established in hydrology, but indeed, we should have been more specific here. As stated above, using power-law regressions on drainage area to infer hydraulic variables across a catchment (or, at least, coarse estimates thereof) is a very common and reliable praxis in hydrology, which is substantiated by a very strong empirical validation. Thus, the goal is not to make a statistical analysis on 4 data points only (which is indeed not recommended), but to anchor the hydrological variables measured in a few sites into these universal scaling relationships (see Figure R2 below on how this indeed works very well). This allows us to have scalable and continuous estimates for any of these variables at any site in the catchment. In theory, one could do this with two data points only and still get good estimates for these variables (even one data point could be enough [Carraro et al., 2020] if the scaling exponent proposed by Leopold and Maddock [1953] were adopted). Hence, having four hydrological stations within a 740-km^2 basin¹ constitutes a very robust dataset for this extent, which matches standards of any hydrological study. An important further advantage of the use of hydrological scaling laws is that one does not need to have many hydrological measures, which enables its application to catchments that are less intensely studied than the one used here.

We took advantage of this Reviewer's comment to notice that the median values of discharge used to derive power-law relationships in the previous manuscript version were considerably lower than the values measured during the sampling period. We therefore derived scaling coefficients based on the mean values of discharge over the period June 11th to 22nd, 2016 (see paragraph reported in the reply above). Mean daily discharge data are freely released by the Swiss Federal Office for the Environment (FOEN) at the link <https://www.hydrodaten.admin.ch/>

Figure R2 below (now also added in the supplement as Supplementary Figure 2) shows the power-law regressions estimated from these discharge values. As one can see, river width and discharge are very well (as expected) correlated to contributing area and show a highly linear relationship on these log-log-plots. As for river depth, we noticed that the depth estimated at the station with lowest contributing area did not follow the expected power-law trend. Presumably, the stage-discharge relationship provided by FOEN for such station is not built on a natural river cross-section, which results in overestimated river

¹ In the previous manuscript version, the basins' surface was mistakenly indicated as 760 km^2 . We corrected this error throughout.

depths. We therefore operated the fit on the three remaining points. Please note: as for velocity, the power law relationship is not fitted to data (which are not available for this variable), but rather inferred from the other relationships, under the hypothesis of rectangular cross-sections, which leads to $v = Q/(wD)$.

Figure R2. Power-law regressions for width, discharge, depth and velocity obtained from values of discharge averaged during the sampling period (June 11th to 22nd, 2016). Dashed vertical lines indicate the range of values of drainage area for the river network used in this study.

Notably, we are highly confident about these trend lines because the exponents of the power laws are very close to those proposed by Leopold and Maddock [1953] ($w \sim Q^{0.5}$, $D \sim Q^{0.4}$, $v \sim Q^{0.1}$) and by Rodriguez-Iturbe and Rinaldo [2001] ($Q \sim A$).

We wish to genuinely thank this Reviewer for prompting us to revise the implementation of the hydrological scaling in much more detail, thereby making the manuscript more accessible to a broad readership and improving the quality of the analyses presented.

#eDNA reads vs taxa concentration/abundance/presence threshold

Line 80-82: *The assumption is the read number is proportional to the eDNA concentration/prediction. Knowing little the relationship between read number and eDNA concentration, I very much like to believe this assumption is reasonable and true. I would appreciate author to provide more justification/citation/literature review.*

R18: This is a good point, and we apologize for being unclear due to the brevity of the text. Please note that we did not assume that read numbers are proportional to eDNA concentration. Rather, we assumed that read numbers follow a geometric distribution whose mean is proportional to the eDNA concentration. Such choice is justified by the fact that high read numbers are generally indicative of high eDNA concentration, although a single read number value is highly affected by uncertainties due to the sequencing procedure. Our choice of a wide distribution for read numbers enables exploiting the quantitative information contained in the read number values, while accounting for their large stochasticity. We added a paragraph in the discussion that better clarifies this aspect (LL. 231-254):

“A key challenge towards a quantitative use of metabarcoding data is expressing the relationship between number of reads and the underlying eDNA concentration for a given taxon. Although several studies [e.g., Evans et al. 2016, Hänfling et al., 2016, Thomsen et al., 2016] found that high number of reads are generally related to high abundances/biomass of species, which is potentially reflected in high eDNA concentrations, a deterministic relationship cannot be found, due to the high stochasticity of read number values resulting from the uncertainties of multiple steps of the eDNA laboratory procedures. For example, different extraction methods can influence diversity results (e.g., Deiner et al. [2015], Spens et al. [2015]), different sequencing platforms have specific error rates when generating DNA sequences (see Laehnemann et al., [2015]), and primer bias can lead to distorted abundance proportions (e.g., Bellemain et al. [2010], Deagle et al. [2014]). However, it has to be noted that the efficiency of the

primer used herein is relatively similar among the three inspected insect orders: an *in silico* evaluation of primer performance showed that the efficiencies of the forward primer for Ephemeroptera, Plecoptera and Trichoptera are 76%, 77% and 80% respectively, while those of the reverse primer are 100%, 100% and 98% respectively [Elbrecht and Leese, 2017]. We expect that recently developed primers [Leese et al., 2020] even more optimized for EPT taxa additionally strengthen the approach proposed here. Furthermore, in our approach the parameter p'_0 , which transforms relative density distributions—proportional to $\exp(\beta^T \mathbf{X}(i))$ —into read numbers, was estimated independently for each genus by the calibration procedure. Hence, possible differential affinities between different eDNA sequences do not affect model results.

In this work, we propose the use of a geometric distribution for read number data from the same sample, whose mean is proportional to the eDNA concentration of the sample. Such choice enables exploiting the quantitative information contained in the read number values, while accounting for their large stochasticity. The choice of such distribution was based on the analysis of the data available for this case study (see Methods), but it would benefit from a validation based on a lab study. Therefore, we call for further research to better elucidate this aspect.”

Line 61-62: local abundance. eDNA is highly criticized for use to represent “abundance.” It is recommended to use to indicate species presence/absence. Similar as above, please provide justification for using eDNA, and the model built with eDNA for taxa abundance.

R19: We agree. Please see our reply above (**R14**). We did not estimate abundances of insects based on eDNA; rather, we assumed in the model that read numbers follow a geometric distribution whose mean is proportional to the eDNA concentration. This then allowed us to infer a match with a species occurrence or not based on presence/absence data. We thus analyze and interpret the final data at the coarser level of presence/absence only. This is now clarified in the text.

#Landscape variables/covariates

The landscape covariates can be local or upstream scale, which make a lot of sense (as eDNA is likely the accumulated results from upstream and local contribution). Line 184-185: The additional “geographical” covariates was not justified. Not sure why this constraint was needed, especially eDNA is results is accumulated from upstream and is accounted in the model.

R20: We have ensured that the justification for the use of geographical covariates is given prominently. As we state in LL. 377-379, “The addition of these ‘geographical’ covariates aimed at allowing eDITH to reproduce spatial patterns uncorrelated to any of the previous covariates (e.g. in the case of a taxon only inhabiting a single tributary of the catchment).” Indeed, in a generalized linear model as the one subsumed by the equation $p_i = p_0 \exp(\beta^T \mathbf{X}(i))$, only spatial patterns of taxon distribution that are proportional (strictly speaking, in the logarithmic scale) to a linear combination of covariates can be reproduced. It is therefore fundamental to include, in the linear predictor, covariates that may potentially mimic the actual taxon distribution. If a taxon is known to be located in a certain subarea, but no environmentally- or physically-based covariates reproducing that pattern are available, the model will not be able to reproduce the data. This prompted us to the inclusion of geographical covariates. It is probably worth underlying that geographical covariates are ‘local’, in the sense that the covariate value at a single node (reach) only depends on the reach itself (namely, on whether or not the reach in question belongs to the corresponding geographical cluster), and not on the upstream portion of the catchment. Indeed, as we report in LL. 375-377, “For each cluster, a corresponding covariate vector was defined with values equal to one for the reaches constituting that cluster, and zero otherwise.”

Model results/interpretation

Line 252: *using an arbitrary 2/3 threshold might not be the best strategy in SDM. Please consider Liu, C., Berry, P.M., Dawson, T.P. and Pearson, R.G. (2005), Selecting thresholds of occurrence in the prediction of species distributions. Ecography, 28: 385-393. doi:10.1111/j.0906-7590.2005.03957.x*

R21: This is a valid point and we thank this Reviewer for this interesting reference on threshold selection. We have now investigated the use of two methods suggested by Liu et al. [2005], namely the prevalence approach (hereafter P) and the average probability/suitability approach (hereafter APS). However, both of these approaches resulted in significantly worse matches between the model and the kicknet data (see detailed explanation in next two paragraphs), indicating that the a-priori chosen 2/3 threshold is better. We also note that the 2/3 threshold was chosen because the same threshold was used by Mächler et al. [2019] for the same dataset. Since the eDITH model is to be seen as a way to translate “upstream-averaged” eDNA measurements into local equivalent eDNA measurements, we deemed appropriate to keep the same criterion already used with the raw eDNA data.

Approach P uses the genus-specific prevalence (fraction of sites occupied) in the training dataset (i.e. the kicknet data, in this case) as threshold. We note that 14 genera detected by eDNA were undetected by kicknet, which would lead to a threshold equal to 0. In this case, all detection probability P_D values for these genera would be converted into presences (as by construction $0 < P_D < 1$), leading to a 0% accuracy for these genera. The opposite case would occur for the genus *Baetis*, always found in the kicknet samples; the corresponding threshold would be equal to 1 and all P_D values would be converted into absences, leading again to a 0% accuracy. Thus, we find this threshold choice not suitable for the type of data we have, as it cannot deal well with the common complete absence of complete presence of a taxon.

In approach APS, the mean P_D across sites is used as threshold. We note that this choice tends to overestimate presences with respect to the fixed-threshold (= 2/3) approach. When we run the analysis with this threshold, such overestimation occurred for 43 out of 50 genera, while for 5 genera the number of presences did not change between the two approaches. This can be explained by the fact that many genera have distributions of P_D across the sampling sites whose means are close to 0 (44% of the genera have mean P_D lower than 0.1). In these cases, many sites are attributed a ‘presence’ value even though the underlying P_D is low. As a result, the mean (across genera) accuracy obtained with the APS method is 78.9%, as opposed to 82.4% with the 2/3-threshold criterion. Since the goal of a threshold criterion is to maximize the performance index (in our case accuracy, which corresponds to OPS—overall precision success—in Liu et al. [2005]), we eventually decided to discard the APS approach and maintain the fixed-threshold approach.

We have added the following paragraph to the Methods section (LL. 454-460): “A number of methods for threshold selection in species distribution models exist in the literature [Liu et al., 2005]; however, preliminary analyses (not reported) showed that methods such as the prevalence approach and the average probability/suitability approach (see Liu et al., [2005]) resulted in lower accuracy estimates (see below) as opposed to the fixed threshold approach. Moreover, the fixed threshold approach $P_{D,i} \geq 2/3$ was used by Mächler et al; since the eDITH model is to be seen as a way to translate “upstream-averaged” eDNA measurements into local equivalent eDNA measurements, it appears appropriate to keep the same criterion already used with the raw eDNA data.”

Line 258-259: *Why do you treat kicknet results as the true presence, and compare modeling result against kicknet? The model was built with eDNA. Are they always consistent with the eDNA sample at the given sites?*

R22: This is an interesting point. Indeed, also the kicknet data will have some errors (mostly false absences). We cannot necessarily assume that kicknet is giving the complete “true” presence, as some

taxa may be missed, and then indeed eDNA can give information on something that is there but was not found in the kicknet (this advantage has been pointed out many times). However, for our approach to be corroborated, we wanted to be as conservative as possible, and use the observed presence in kicknet as the reference. Having sampled and analyzed all these kicknet samples ourselves, we can be of very high certainty that the level of false presences is very low or even zero. This makes it the most robust but also most conservative point to compare with eDNA data.

Moreover, we compared model results with kicknet because the latter is a totally independent dataset with respect to the eDNA data, with which the model was calibrated. We are aware that kicknet-based presence/absence (p/a) is likely biased due to e.g. elusive genera, and this is why we also provided figures on average accuracy of our model when false positives are treated as good estimates. The comparison between eDNA and kicknet data is already carried out (with different tools) in Mächler et al. [2019]. If we evaluate the accuracy between eDNA data and kicknet data, we obtain results that are akin to the ones here reported: for 12 genera, eDITH-based p/a is more accurate than eDNA data in reproducing the kicknet p/a; for 25 genera the vice versa is true; for the remaining 13 genera, both model and eDNA yield the same accuracy. Indeed, if for instance in a certain tributary a given taxon was found by kicknet but not by eDNA, the eDITH model, being trained on the eDNA data, will not be able to predict presence for that taxon in that branch. As a consequence, it is reasonable to expect that the model does not perform much better than the eDNA measurements alone with respect to accuracy with the kicknet data.

Finally, we note that it would be formally wrong to evaluate the accuracy of model results in terms of local p/a with respect to the eDNA data. Indeed, the latter is an aggregated measure of the upstream taxon distribution while the former refers to a local variable (because upstream contributions have been disentangled by the model). The same reasoning can be applied with respect to the comparison between kicknet-based (local) and eDNA-based (upstream) p/a. Conversely, the measure of “consistency” between model results and eDNA data that we proposed is the goodness-of-fit test that compares quantities of the same type (i.e. expected read number values predicted by the model with observed read numbers). Such test showed a remarkable consistency between model outputs and metabarcoding data since, for all genera, the fraction of sites where the test gave a positive result is higher than 90% (see Fig. 7b).

We now added the following paragraph to the Methods section (LL. 473-479): *“Note that calculating the accuracy between model predictions and eDNA data with the method used for the model vs. kicknet comparison would be formally wrong: indeed, eDNA data are an aggregate measure of the upstream taxon distribution, while eDITH-based presence estimates refer to a local variable, because upstream contributions have been disentangled by the model. Conversely, an adequate measure of consistency between model predictions and eDNA data must compare quantities of the same type (i.e. both referred to the upstream catchment). To this end, we utilized the goodness-of-fit test detailed below.”*

Detailed comments attached

R23: In the following we copied and replied to detailed comments in the annotated PDF document provided by the reviewer that did not overlap with the previous points.

L. 17: “Highly resolved” What do you mean?

R24: We rephrased as *“at high spatiotemporal resolution”*.

L. 74: The threshold of reads to considering present was not described.

R25: No threshold on read numbers was adopted. All genera with at least one positive read number at one site were included in the model. We did so to be most parsimonious (as any threshold application would require additional justifications). The approach chosen is the most conservative. Please see extensive reply to Reviewer #1 on the same subject (**R6**).

L. 100: It would be even better, if you go back to those hot spots (ovals) streams to verify the high Genus richness by eDNA samples or kicknet sample.

R26: This is a good suggestion. However, the amount of field work would currently go beyond our abilities (also because all such field work at our institutions has been suspended for COVID-19). Importantly, however, we have corroborated the plausibility of these hotspots with naturalist experts.

L. 105 "estimating characteristic decay times of eDNA". This points needs to further explain on how and why.

R27: Decay time is estimated as a free parameter of the model by means of an Adaptive Metropolis algorithm. Please refer to the reply (**R40**) to Reviewer #3's remark on the same subject for a more detailed argumentation on the interpretation of the value of decay time.

L. 110: "How did you exclude the false positive?"

R28: In a first round, we assessed the accuracy of the model as the fraction of true positives and true negatives over the whole number of tests (= 60 sites). This standard definition of accuracy mirrors that of OPS (overall prediction success) of Liu et al. [2005]. In a second round, we also included false positives as possible successful predictions, justified by the fact that no capture of EPT taxa in a certain area does not necessarily imply their absence.

L. 115: How many replicates of eDNA samples?and where in a stream "cross-section" that you sampled?

R29: Three independent water samples were taken at each site and separately analyzed, resulting in three true replicates, which we termed in the manuscript as 'replicates'. Water samples were taken close to the river banks. We included such information in the Methods section (LL. 285-288).

L. 121 this doesn't explain why the model underpredict the EPT. L. 122: This is not clear. You claimed your model considered the upstream biodiversity, because the stream longitudinal connectivity. But now you said the downstream solely based on downstream sampling.

R30: Based on this and on comments from all other Reviewers, we now expanded this analysis and revised the whole paragraph. Please see detailed answer to Reviewer #1 on this subject (**R3**).

L. 125 "corroborate"?

R31: We wanted to show that our results hold irrespective of the choice of sites on which the eDITH model is calibrated. We have revisited that sentence and think that our wording is correct.

L. 130: *Down the road, it might be worth a while to explore how many sites are needed using eDNA sample. (similar research was done for kicknet or cross-validation bias). Elith, J., Leathwick, J.R. and Hastie, T. (2008), A working guide to boosted regression trees. Journal of Animal Ecology, 77: 802-813. doi:10.1111/j.1365-2656.2008.01390.x*

R32: This is definitely a key research question for future eDNA studies. We are actually currently developing a simulation study where we use the eDITH model to assess what is the optimal sampling strategy for an eDNA campaign in a river network. We have now included a sentence that underlines the need for such research direction (see reply **R3** to Reviewer #1).

L. 132: *"but its results consist in a number of pointwise estimates that could hardly be projected into biodiversity maps" why not?*

R33: It would be indeed possible to use pointwise estimates of eDNA (not model)-based biodiversity to fit statistical models and obtain interpolated values of biodiversity along stream directions. Refined techniques for this task exist (e.g. the SSN R-package [Ver Hoef et al., 2014]). However, such approaches would completely discard the processes of transport and decay of eDNA, which make eDNA measurements aggregated indicators of the upstream biodiversity. For the sake of clarity, we now modified the sentence as *"biodiversity maps at high spatial resolution"*.

L. 158: *"the channelized portion of the drainage network". All stream (concrete) channelized? Did the GIS formed stream lines agree with the actual stream (by using the 0.5km² threshold)?*

R34: We apologize for this misunderstanding. The streams studied were not concrete-channelized, but mostly in a relatively natural state with some river bank modifications, but without any concrete. We had used "channelized" *sensu* O'Callaghan and Mark [1984], namely as perennial reaches, *"at which runoff is sufficiently concentrated that fluvial processes dominate over slope processes"* [O'Callaghan and Mark, 1984]. We now modified our sentence as (LL. 327-328) *"the channelized (i.e. perennial, see O'Callaghan and Mark [1984]) portion of the drainage network"*. We performed a qualitative comparison between the extracted network and the vectorial hydrographic network provided by the Swiss Federal Office of Topography, which we deemed satisfactory. We acknowledge that many small headwater reaches were not captured by the adoption of a threshold area of 0.5 km², which explains why the stream order values for this catchment in this study range from 1 to 5, whereas they are up to 7 in Mächler et al., [2019]². We refrained from using a smaller value of threshold area because of the reason outlined in our reply after next.

L. 160: *It will be more clear if you describe your definition of reach before saying how many of them. The definition of the reach matters a lot, as you are using that as the basic spatial unit in your modeling framework.*

R35: Thanks for this suggestion. We reverted the order of the two sentences in the new manuscript version.

L. 164 *"A trade-off between the contrasting requirements of likeness between extracted and real river networks..." what do you mean here? It's not clear. what trade-off you made? What did you end up sacrifice ?*

² We already commented on this fact in the caption of Extended Data Figure 4 (now Fig. 1c).

R36: We apologize if this was not clear. A higher threshold area value would have resulted in less reaches (given the definition of a reach that we provide in the text), thereby providing a poorer representation of the river network, with missing information from most headwater streams. Conversely, the number of reaches has a substantial impact on the computational time required. Note that the current simulations for 50 genera and the scenario where all sampling sites are used to calibrate the model take around 4 days to complete, while the additional simulations based on subsets of the 61 sampling sites take more than a week. From a computational viewpoint, it would have been problematic to further increase the resolution of the river network.

We now modified the sentence as (LL. 332-337) *“Note that a lower threshold area would have resulted in a higher number of reaches, implying a more refined discretization of the river network but also an increased computational burden for the subsequent model runs. Hence, we chose the highest value of threshold area such that: i) the extracted river network retained all reaches where sampling sites were located; ii) a qualitative comparison with the vectorial hydrographic network provided by Swisstopo resulted adequate.”*

L. 177: “These covariates can either reflect local or upstream characteristics” Yes! However, why did you choose some in local vs upstream (in land use covariate)?

R37: We chose to compute geological covariates as upstream values because different geological classes can have different effects on the chemical composition of streamflow (see also Carraro et al., [2017]), which in turn can affect the taxon’s affinity for the local habitat. In this context, it is clearly important to consider the whole area upstream of a site when calculating the covariate value at that site. Conversely, land cover covariates were considered as potential drivers of local suitability, and were therefore calculated as local values (because no streamflow dynamics are involved). It would certainly be possible to add local geological covariates and upstream land cover covariates to this model; however, this would further complicate the estimation of parameters and would likely cause multicollinearity in the linear predictor matrix (since, for instance, in headwater streams local and upstream covariate values coincide). We now describe this reasoning in the revised version of the manuscript (LL. 366-371): *“These covariates can either reflect local or upstream characteristics. Land cover covariates were evaluated as local values, because they are assumed to potentially have a role in determining local taxon suitability (see also Kaelin and Altermatt [2016]). Geological covariates were calculated as upstream averaged values, because they are likely to affect the chemical composition of streamflow at a site; such process could affect local habitat suitability but is driven by the upstream catchment [Carraro et al., 2017].”*

Hope this helps. Thanks for the opportunity to review the paper and provide the feedback/comments.

R38: We wish to thank this Reviewer for her/his insightful, detailed and constructive comments which very much helped improve our manuscript.

Reviewer #3 (Remarks to the Author):

My overall comment is that the work will be of broad interest, but that the presentation is so brief as to be incomplete. This makes the work very difficult to evaluate. Simply put, the work seems very good and topic very interesting, but the article is too short and lacks necessary details. If a word limit is imposed by the journal and can not be modified, then I suggest another outlet would be more suitable. Most of comments, as you will below, have to do with there not being enough information, or space to even suggest some alternative explanations, hypotheses, or considerations.

R39: We thank this Reviewer very much for her/his enthusiastic and supportive feedback. We had initially written the manuscript to be relatively short, but then realized, also based on the Reviewers' comments, that we should rather be much more extensive. In the present version, we considerably expanded on the details and assumptions of the model, as well as on the perspectives of our approach. The whole manuscript is now about twice as long, and with several figures added to the main text and/or supplement. This also includes several specific explanations added based on specific requests by any of the three reviewers. We are confident that we could give all the details and discussions required.

Line 51: It is not clear the role of hydrology in the paper and in the model. Deposition is a very important factor in particle (i.e., non-solute) transport, and it is not clear whether this is dealt with in your model here, or whether previous work has investigated the role of deposition and can quantify it or rule it out as negligible. This section needs some more detail aside from a few (limited) factors listed in parentheses.

R40: We thank this Reviewer for raising this point. We have now added substantial information on the hydrological components of the model (LL. 340-363, see replies **R16-R17** to Reviewer #2), and we specifically discuss the individual hydrological variables in more detail. Indeed, removal of eDNA has different components, such as decay or deposition. In our model, both processes are subsumed in the parameter τ . We did not explicitly model deposition independently, because i) there are very few empirical data on it; ii) it would have led to a confusion effect in model estimates. We have now better clarified this aspect in the manuscript (LL. 398-404): “Note that Eq. (1) does not explicitly account for deposition of eDNA particles on the river bed. Decay and deposition are both relevant processes affecting eDNA removal, and thus detection, in stream water [see e.g. Shogren et al., 2017]. We chose not to model these processes independently because it would be hardly possible to disentangle the roles of degradation of genetic material and that of gravity-induced deposition. Decay time τ is therefore to be seen as a characteristic time (linked to half-life $T_{1/2}$ by the relationship $T_{1/2} = \ln(2) \tau$) for the presence of eDNA in water, regardless of the source of its depletion”.

Line 54 (also line 75) The status of the model is unclear throughout. Is it an existing model or is it being newly described here? The text gives the impression that it exists and is fully described in ref. 25; however, the methods section spends a great deal of time describing it. I was not sure whether I am peer-reviewing the model itself or whether it has been reviewed and I am reviewing its application to eDNA. I think a few lines of explanation of the model and its history/previous applications would help here.

R41: We apologize for this not being totally clear. We have now clarified this aspect in the manuscript. The model here presented builds upon that of Carraro et al., [2017, 2018]. While Carraro et al. [2018] developed the formal principle of the model and applied it to qPCR data for two species, we here for the first time expanded it for use with next generation sequencing data, showed its validity on a large-scale, multispecies eDNA dataset, and compared its ability to assess biodiversity with classic approaches. In fact, Carraro et al. [2018] developed some basic elements concerning transport and decay of eDNA (corresponding to Eq. (1) of the manuscript), but did not assess presence/absence maps based on maps of

relative density estimated for the target species. Thus, major parts of the model are new, and require such an extensive description, while the mathematical core of the model has been already derived previously. We now better describe the historic context of the model, and its advances in our study. Importantly, the journal's requirements ask to provide all material needed to reproduce the work in the methods sections. Thus, we also iterate on elements that had been developed, but are crucial for this study.

We specifically say that Eq. (1) is taken from the aforementioned publication (see L. 388: "*The eDNA transport component of the eDITH model is derived from Carraro et al. [2018]*"), while LL. 405-453 ("*We here further hypothesize...*") describe additional aspects and extensions that lead to the eDITH modelling framework described and developed in this manuscript. For the sake of clarity, we revised the sentence as follows (LL. 59-60): "*Here, we develop an integrated hydrology-based modelling framework (hereafter termed eDITH—eDNA Integrating Transport and Hydrology), built on the approach of Carraro et al., [2017, 2018]*".

Line 60: How were these maps produced?

R42: These maps were produced with custom-made functions based on publicly available GIS datasets. Methodologically, the procedure is outlined in the methods section of the manuscript. We double checked that the procedure is described in detail. From a practical viewpoint, custom MATLAB scripts were used to analyze the data and produce figures. Note that all codes and scripts are publicly available. We now added a sentence on code availability: "*MATLAB scripts reproducing the results of this manuscript are available at <https://github.com/lucarraro/eDITH-thur>*".

Line 61: awkward wording: perhaps "of each taxon's" of "of the presence of each taxon"

R43: Thanks. We reworded accordingly.

Line 63: These samples are not described. How many sites? At first it looks like 61 sites (line 65) but later only 60 sites (line 257). Are the same kick-net samples used for both purposes? Again, a bit more text would help here. The citation is for the methodology, but as used here looks like these data are published. Please clarify that these are also newly collected data.

R44: We apologize if this aspect was unclear. We have now added some more specific information on the samples. In total 61 sites were studied. At 60 of these sites, eDNA and kicknet samples were taken. At one site, only eDNA samples were taken, while the kicknet sample was lost due to handling error.

We have also added additional text on the kicknet sampling to the method section. For the sake of clarity, we moved the reference to Barbour et al., [1999] after "standardized" and rephrased (also incorporating the next Reviewer's comment) as (LL. 73-77) "*Three independently replicated eDNA samples and a pooled benthic invertebrate kicknet sample were taken at each of 61 sites (60 for kicknet) across the Thur catchment in Switzerland (Fig. 1) in June 2016. Site selection, data collection and technical processing of samples for both eDNA and kicknet are described in the Methods section and in Mächler et al. [2019], the original publication of the presented sequencing data.*"

Line 65: Change "Threefold replicated" to "Three eDNA and three invertebrate..." These are, at best, technical replicates.

R45: We revisited the text and corrected (see comment above). For the sake of clarity, we also revised the Methods section (LL. 273-291):

“In June 2016, diversity data were collected at 61 sites in a 740 km² sub-catchment of the river Thur, northeastern Switzerland. No singular rain events took place during the sampling days. At each site, a standardized [Barbour et al., 1999] three-minute kicknet sampling applied to three microhabitats was performed to collect benthic macroinvertebrates. The subsamples from the three microhabitats were pooled and stored in ethanol. In the laboratory, debris was removed and all individuals belonging to may-, stone-, and caddisflies (Ephemeroptera, Plecoptera, and Trichoptera, abbreviated as EPT) were identified under a stereomicroscope to the genus or species level if applicable. One sample was lost due to handling error, and thus we subsequently only had EPT kicknet sample data from 60 sites.

At the same 61 sites at which kicknet data were collected, also eDNA samples were taken. We collected three independent samples of 250 mL of river water at each sites (sampled below the surface and well above the river bottom). In small streams (up to about 1-m wide), the water was taken from the middle of the stream, while in larger streams the water was taken about 0.5 m away from the shore side. All three water samples were filtered on site on separate GF/F filters (pore size 0.7 µm Whatman International Ltd.), which were stored on ice immediately after filtering, and frozen at -20 °C within a few hours. Subsequently, these three samples were analyzed separately and as independent replicates”.

Line 67: I find this too brief, even in the method section there are too many details missing.

R46: We have now substantially expanded the Methods section of the manuscript. Specifically, we added more details concerning the eDNA extraction and processes (LL. 299-304): *“In short, three PCR replicates were performed for each eDNA replicate with primers that contained an Illumina adaptor-specific tail, a heterogeneity spacer and the amplicon target site. These three PCR replicates per sample replicate were pooled and indexed in a second PCR with the Nextera XT Index Kit v2 (Illumina). We measured the concentration of all indexed PCR reactions and pooled them in equimolar parts to a final library, which we ran twice on two consecutive Illumina MiSeq runs to increase sequencing depth”.* See also reply **R57**.

Line 72: I would like to see some evaluation of whether this is a suitable read depth. What does “pooled over genera and replicates” mean? How were data handled i.e. if a genus was found with 2 reads in only one “replicate”, 10 reads in two replicates, etc(?)

R47: We are very confident that we have sufficient read depth. Specifically, we have sequenced the library twofold, in order to increase read depth. Moreover, as shown in Mächler et al. [2020], the taxa observed in this sequencing dataset are generally saturating. We acknowledge that the primers used are targeting eukaryotic diversity in general and were not only specific to EPT taxa, which might lead to an underestimation of detections of EPT by eDNA (i.e., false absences). This, however, is conservative with respect to our findings, as we would miss taxa with eDNA where the kicknet sample indicated their presence. We now better explain this in the manuscript (LL. 318-322).

The OTUs found in this dataset are generally saturating [Mächler et al., 2020]. We acknowledge that the primers used are targeting eukaryotic diversity in general and were not only specific to EPT taxa, which might lead to an underestimation of detections of EPT by eDNA (i.e., false absences). This, however, is conservative with respect to our findings, as we would miss taxa with eDNA where the kicknet sample indicated their presence.

As for the second part of the question, we underline that all read number values (61 sites * 3 replicates * 50 genera) were used in the subsequent calculations; in particular, read number values equal to 0 were

also retained. The median number of reads per site, pooled over genera and replicates, is the median of a data set containing 61 components (one per site), each of which corresponds to the sum of read numbers across 150 values (50 genera and 3 replicates) for that particular site. We have expanded on this in the manuscript (see reply **R6** to Reviewer #1).

Line 77: What is the source of the species distribution model? Is it constructed from the kicknet sample results?

R48: We have added some clarification on this. The species distribution model indicated in L. 77 in the previous manuscript version refers to the equation reported in L. 202 in the previous manuscript version $p_i = p_0 \exp(\beta^T X(i))$, which links the relative density of a taxon to a vector of covariates. We clarified this aspect in the Method section (LL. 397-398): “Eq. (2) represents a generalized linear model, a widely used class of species distribution models [Guisan et al., 2017]”.

Line 79: Please give some detail regarding the “uncertainties”. I did not find it obvious which part of the methods what this was referring to. There are many that are never dealt with (PCR bias, contamination, ...). At a minimum these could be mentioned in a discussion.

R49: Also prompted by a comment from a Reviewer #2 (**R18**), we now included a paragraph in the discussion where we discuss possible uncertainties commonly inherent to PCR and the laboratory procedure. This could affect read numbers and eDNA concentration, but would be again conservative with respect to our conclusions, and we discuss how we accounted for that in our approach.

Line 104: Please define “biodiversity in general” more specifically - this refers to the fig.2 covariate analysis which is not mentioned anywhere except in the methods.

R50: For the sake of clarity, we now rephrased the sentence as (LL. 93-94): “we were able to assess the role of environmental covariates in driving the spatial distribution of single taxa (Fig. 2a) and EPT biodiversity (Fig. 2bc) across the study catchment”. Fig. 2 is now mentioned 7 times along the manuscript.

Line 113: Can the authors rule out DNA from upstream sources, or cross-contamination of samples as alternative explanations here(?) Alternative explanations are rarely offered in the text.

R51: We can rule out cross-contaminations to a very high certainty, as we had used appropriate negative and blank controls in our laboratory procedure. We have a state-of-the-art eDNA laboratory, and have also been strongly involved in setting standards for this type of work (e.g., Deiner et al. [2015], Blackman et al. [2019], Pawlowski et al. [2018, 2020]). We have no indications of any systematic or prevalent cross-contaminations. Moreover, cross-contaminations would be expected to create noise in the dataset, and not patterns. We now included this sentence in the Methods section (LL. 291-293): “To prevent cross-contamination, the eDNA samples were collected a few meters upstream of the kicknet sampling, and kicknet sampling and eDNA sampling were performed by different people.” As for the DNA transport from upstream sources, we—as described in the model section in detail—consider this as a specific and inherent aspect of eDNA, which we turn to our benefit using hydrological modeling approaches. Thus, eDNA is indeed possibly released at upstream areas, and by using appropriate modelling, we can account for its transportation and decay. By this, we can infer the relative contributions of local eDNA production vs. transport from upstream, which allows us to then predict the occurrence of a taxon in space.

Line 115: Please give criteria for a "match" and add this to fig legend.

R52: We did so. We now performed two-sample Kolmogorov-Smirnov tests to substantiate our claims. The amended paragraph reads (LL. 114-123): *"As shown in Fig. 6, the distribution of genus richness predicted by the model in the headwaters (reaches of stream order 1) matched the one assessed via the kicknet sampling: a two-sample Kolmogorov-Smirnov (2KS) test did not reject the null hypothesis that the two samples come from the same distribution ($p = 0.43$). Conversely, eDNA data alone underestimated genus richness in low stream order reaches; a 2KS test between eDNA and kicknet richness at reaches with stream order 1 yielded $p < 0.001$. This is presumably due to the low number of replicates per single location and the high heterogeneity among these. Instead, the EPT biodiversity predicted by the model at the high (≥ 4) stream order reaches was lower than the one measured by the kicknet dataset (2KS test: $p = 0.002$), whereas in these reaches the genus richness based on eDNA data matched the kicknet-based richness (2KS test: $p = 0.98$)."*

Line 116: Is there a known relationship? If so please elaborate.

R53: It is what we actually observe by comparing eDNA data and kicknet. We are not aware of previous studies performing such comparisons (except for Mächler et al., [2019], based on the same dataset).

Line 119-124: Are there alternative explanations? This section is not easy to understand without any explanation of the model itself, which is mostly in "methods". Why and how are downstream sites more prone to "error" - does error mean false positives, negatives, both?

R54: We have now added some more explanation to this section. The journal's format, however, requires the Methods section to be located at the end of the manuscript, which thus needs to be referred to. Downstream sites are more prone to false negatives because the eDNA signal at a downstream site can likely be interpreted as coming from an upstream source, rather than originated locally. We have now expanded this paragraph (LL. 127-130): *"On the contrary, model predictions at the downstream portion of the catchment are solely based on information from downstream sampling sites, and are thereby more prone to error, since the model might interpret an eDNA signal detected at a downstream site as originated from an upstream source rather than locally."* See also the detailed reply to Reviewer #1 on this subject (**R3**).

Line 136: What are these first principles? At a minimum, a short description of the model and how it works with eDNA data (and why kick-net data would not be suitable). Why not consider the fact that kick-net samples might also have individuals that have drifted from upstream, and may not consist of the local community. I am missing some ecological considerations of the approach and its potential limitations.

R55: The "hydrological first principles" referred to in the manuscript are the use of scaling relationships linking width, depth, discharge and velocity to drainage area. Please see our detailed reply to Reviewer #2 on the same subject (**R16**). We have now added substantial text in this respect. We apologize if we had been a bit short on these descriptions. Please note that relationships are very well supported and long-studied in hydrology, and have thus a very high robustness. Note also that our approach is a model of catchment-scale transport of a tracer that is passively advected by stream water, and the transport distances are at the order of a few kilometers. The assumption of passive transport is (with due approximations) suitable for environmental DNA (as also shown by other studies, e.g. Deiner & Altermatt [2014], Pont et al., [2018]).

For the second part of the comment, namely the possibility of drift in stream invertebrates, we can exclude this to a very high likelihood. Stream invertebrates can indeed be drifted; however, the scale of

this phenomenon is generally a few meters of up to tens of meters, as has been already well established in freshwater ecology since the Seventies, especially by seminal work by Elliott [1971, 2002, 2003]. Thus, the kicknet samples are giving a very local description of communities; the scale of few tens of meters within which drifting might occur is such a fine-grained resolution that it can be ignored by our model. Also, we note that we took very high precautions for not cross-contaminating samples, and have washed and cleaned the kicknet between different sampling sites. False positives are thus virtually impossible to have in the kicknet approach. We now mention this in more detail (see also reply **R4** to Reviewer #1).

Line 148: Please give details of the filters and extraction methods.

R56: We have now added these details in the Methods section (see extensive reply **R15** to Reviewer #2).

Line 152: If this pipeline is not published or described, then it is impossible to evaluate the approach. Please provide at least minimal information on the analysis. I would be interested in how the multiple samples ("replicates") were handled - did an OTU (or ASV?) need to be in at least two to be considered present? etc.

R57: The bioinformatical pipeline used herein is actually published and openly accessible. It is extensively described in Mächler et al. [2019]. For space reasons we refrain from describing the bioinformatical pipeline again, but based on this comment we now added (LL. 304-316) a short summary of the pipeline, and refer to the extensive version:

"Raw data were demultiplexed and read quality was checked with FastQC [Andrews et al., 2010]. Thereafter, end-trimming (usearch, version 10.0.240) and merging (Flash, version 1.2.11) of raw reads were performed, before primer sites were removed (cutadapt, version 1.12) and reads were quality-filtered (prinseq-lite, version 0.20.4). Next, we used UNOISE3 [Edgar, 2016], which has a built-in error correction to reduce the influence of sequencing errors, to determine amplicon sequence variants (ZOTUs). To reduce sequence diversity, we implemented an additional clustering at 99% sequence identity. We checked ZOTUs for stop codons of the invertebrate mitochondrial code, to ensure an intact open reading frame. For the taxonomic assignment, all COI-related sequences were downloaded from NCBI and ZOTUs were blasted against the NCBI COI collection. We extracted the top five best blast hits and then used the R packages "taxize" [Chamberlain and Szöcs, 2013] (version 0.9.7) and "rentrez" [Winter, 2017] (version 1.2.2) to acquire taxonomic labels. We modified the selected COI sequences with the taxonomic labels in order to index the database. The ZOTUs were then assigned to taxa using Sintax (usearch) and the NCBI COI-based reference."

The replicates were treated as independent samples and did not influence the bioinformatic analysis of the ZOTUs. However, one of the implemented steps (UNOISE3) excludes ZOTUs with less than four reads in the total data set from the further analysis by default, as low-abundance ZOTUs are more susceptible to contain errors reproduced by chance or bias [Edgar, 2016].

Line 168: What is the time interval for which the median is calculated? Annual? Are four stations enough to adequately characterize the entire catchment? Some discussion of potential limitations are needed here and elsewhere.

R58: Medians were calculated from discharge duration curves spanning more than 30 years of data. However, we have now re-run all analyses based on mean values of discharge during the sampling period (June 11 to 22, 2016). Please see replies to Reviewer #2 (**R16-R17**) in this respect.

In short, using power-law relationships to characterize hydrological variables across a catchment is a long-established method in hydrology [Leopold and Maddock, 1953; Rodriguez-Iturbe and Rinaldo, 2001]. The four stations are indeed sufficient to characterize the entire catchment, because they are positioned at very different stream sizes, and thus are very well suited to parametrize the relationships which are governed by universal scaling laws. Indeed, the fit of the power law relationships to the observed values of width, discharge and depth is very satisfactory (see Figure R3 below—now added in the supplement as Supplementary Figure 2), and, remarkably, the scaling exponents found for the four relationships are very similar to the values proposed by Leopold and Maddock [1953].

Figure R3: Scaling of hydrological variables (same as Figure R2)

Moreover, we also included a sentence in the discussion on uncertainties and limits related to the hydrological model (LL. 205-216): *“The eDITH model relies on estimates of relevant hydrological variables such as water velocity and discharge across the catchment. In the case study here presented, we obtained these data from power-law regressions based on four hydrological stations. This approach allows capturing the essential features of hydrological transport and hydrological dynamics at large scales, as studied herein, and is supported by longstanding evidence in hydrology and geomorphology [Leopold and Maddock, 1953; Leopold et al., 1964; Rodriguez-Iturbe and Rinaldo, 2001]. Importantly, all uncertainties associated with the processes of eDNA shedding, transportation, decay, extraction, and bioinformatic processing (all of which are coarsely accounted for in the eDITH approach) are at a scale that a more elaborate hydrological model would not significantly improve the predictions of taxa distribution and biodiversity. The use of universal hydrological relationships also allows the application of our approach to riverine systems with scarce hydrological data, thereby enabling monitoring of highly biodiverse but hardly accessible ecological systems [Grill et al., 2019].”*

Line 176: Seems to only be in the legend of Fig. 2 and does not add anything to the study as presented. My apologies if i missed it in the text other than line 104-5.

R59: We now added a paragraph (see reply **R2** to Reviewer #1) on distribution maps for some EPT genera and the respective role of covariate. Fig 2 is thereby necessary for the understanding of this new paragraph.

Line 201: How are these densities estimated for each taxon at each site? From kicknet samples? Are these quantitative enough? Are there other data available?

R60: This may be a misunderstanding. Relative densities p_i are estimated via the eDITH model and not directly compared to the kicknet data. Basically, for each taxon we fit the model on triplets of

corresponding read number data measured at the various sites, and find the spatial distribution of p_i (together with an estimate of decay time τ) that best explains the pattern of read numbers observed. The comparison with kicknet data for each taxon is then made on a transformed variable, namely the presence/absence map that is derived from p_i via the approach described in LL. 438-466 (LL. 238-259 of the previous version). Historical data for EPT taxa on the Thur catchment are available from the CSCF (Centre Suisse de Cartographie de la Faune) database and were analyzed in Mächler et al., [2019]. We did not include them in our model because these data are not spatially distributed within the catchment. We have now revisited the text to ensure this is all clear.

Line 123: This is an interesting approach (geometric distribution). Has it been used elsewhere? Was there any test as to whether it provided a better empirical fit than other distributions? Does the bioinformatic analysis pipeline need to consider this?

R61: The geometric distribution has been chosen for the reasons reported in LL. 415-418 (LL. 215-218 of the previous version). In a preliminary phase, the negative binomial and the Poisson distribution were also tested. However, the negative binomial distribution depends on two parameters, which hinders the estimation of other parameters; the Poisson distribution, as stated in the text, does not allow accounting for fat tails (large read numbers) as opposed to the geometric distribution. Note that, however, three values are likely too few to allow a systematic assessment of the fit of different distributions. Therefore, our choice is to be seen as a working hypothesis, which provides a reliable interpretation of the data (as highlighted by the goodness-of-fit data histogram of Fig. 7b). It would indeed be interesting to test our hypothesis on larger datasets, with e.g. at least 10 read number values extracted from samples taken at the same site. As for possible links with the bioinformatics pipeline, note that our approach does not pertain to the detection of eDNA sequences, but rather to the biological interpretation of the output of the bioinformatics pipeline (i.e. read numbers).

Line 265: There is a lot of important on the read numbers - the paper would benefit from some discussion as to how this might be problematic. See my comment re source of abundance data (above, comment re line 201).

R62: Also prompted by a comment (**R18**) from Reviewer #2, we added a paragraph (LL. 231-254) in the discussion where we comment on the uncertainties related to the read number values, and how our assumption of geometrically distributed read numbers sounds plausible but requires further validation in the lab.

Line 314: How old?

R63: We have now added a paragraph to the discussion, in which we discuss our findings to the historic dataset, and have added an appropriate reference to the Swiss Data Centre on faunistic data (CSCF).

Line 335: I could not see the cross in my copy.

R64: Thanks for noticing this. We increased the size of the cross.

Line 338: Fig. 2a please use a more obvious y-axis title(?) At a minimum, state in the legend what H0 is. I would also try to add the genus names, to have a bit of ecology in the figure. Were there any patterns e.g. were the less

accurate taxa always E, P, or T? Perhaps members of a family? As presented, the fig is not stand-alone. Fig. 2b,c is not helpful unless covariates are identified. I propose removing one panel and writing out the covariates. The panels are not adequately described in the text, and could even be removed.

R65: Thanks for the suggestions. We now recast all figures of the manuscript, including those that were previously presented as Extended Data. Note that results have minimally changed since all models were re-run (see reply **R17** to Reviewer #2). We now included in the caption a definition for H_0 . We refrained from introducing genus names and covariate names in the figures, as they would have become more cumbersome. We did not observe any relevant pattern in terms of differences in accuracy between the three different orders. We now refer to the Fig. 2 in the newly added paragraph (see reply **R2** to Reviewer #1) which comments upon the distribution of some relevant genera as predicted by the model.

Again, many thanks for these detailed and constructive comments, which improved our manuscript.

References used in responses to the three reviews

- Andrews, S., Krueger, F., Segonds-Pichon, A., Biggins, L., Krueger, C., and Wingett, S. (2010) FastQC: a quality control tool for high throughput sequence data. Available at <http://www.bioinformatics.babraham.ac.uk/projects/fastqc>.
- Barbour, M. T., Gerritsen, J., Snyder, B. D. and Stribling, J. B. (1999) Rapid bioassessment protocols for use in streams and wadeable rivers (USEPA, Washington D.C.)
- Bellemain, E., Carlsen, T., Brochmann, C., Coissac, E., Taberlet, P. and Kausarud, H. (2010) ITS as an environmental DNA barcode for fungi: an in silico approach reveals potential PCR biases, *BMC microbiology* 10(1), 189.
- Blackman R. C., et al. (2019) Advancing the use of molecular methods for routine freshwater macroinvertebrate biomonitoring – the need for calibration experiments. *Metabarcoding and Metagenomics*, 3: e34735.
- Buchner, D., Beermann, A. J., Laini, A., Rolauuffs, P., Vitecek, S., Hering, D., and Leese, F. (2019) Analysis of 13,312 benthic invertebrate samples from German streams reveals minor deviations in ecological status class between abundance and presence/absence data. *PLoS ONE*, 14(12).
- Carraro, L., Hartikainen, H., Jokela, J., Bertuzzo, E. and Rinaldo, A. (2018) Estimating species distribution and abundance in river networks using environmental DNA. *Proceedings of the National Academy of Sciences of the United States of America* 115.
- Carraro, L. et al. (2017) Integrated field, laboratory, and theoretical study of PKD spread in a Swiss prealpine river. *Proceedings of the National Academy of Sciences of the United States of America* 114.
- Carraro, L., Toffolon, M., Rinaldo, A., and Bertuzzo, E. (2020) SESTET: A spatially explicit stream temperature model based on equilibrium temperature. *Hydrological Processes*, 34(2), 355-369
- Chamberlain, S. A. and Szöcs, E. (2013) taxize: taxonomic search and retrieval in R. *F1000Research* 2, 191.
- CSCF (2020) Info fauna - Swiss topic center on fauna, Available at <https://lepus.unine.ch/cartof/> (2020).
- Deagle, B. E., Jarman, S. N., Coissac, E., Pompanon, F., and Taberlet, P. (2014) DNA metabarcoding and the cytochrome c oxidase subunit I marker: not a perfect match. *Biology letters*, 10(9).

- Deiner, K., and Altermatt, F. (2014) Transport distance of invertebrate environmental DNA in a natural river. *PLoS ONE*, 9(2).
- Deiner, K., Walser, J. C., Mächler, E., and Altermatt, F. (2015) Choice of capture and extraction methods affect detection of freshwater biodiversity from environmental DNA. *Biological Conservation*, 183, 53-63.
- Edgar, R. C. (2016) UNOISE2: improved error-correction for Illumina 16S and ITS amplicon sequencing. *bioRxiv*.
- Elbrecht, V., and Leese, F. (2017) Validation and development of COI metabarcoding primers for freshwater macroinvertebrate bioassessment. *Frontiers in Environmental Science*, 5, 11.
- Evans, N. T., et al. (2016) Quantification of mesocosm fish and amphibian species diversity via environmental DNA metabarcoding. *Molecular Ecology Resources*, 16, 29–41.
- Elliott J. M. (1971) The distances travelled by drifting invertebrates in a Lake District stream. *Oecologia*, 6(4): 350-379.
- Elliott J. M. (2002) A continuous study of the total drift of freshwater shrimps, *Gammarus pulex*, in a small stony stream in the English Lake District. *Freshwater Biology*, 47(1): 75-86.
- Elliott J. M. (2003) A comparative study of the dispersal of 10 species of stream invertebrates. *Freshwater Biology*, 48(9): 1652-1668.
- Grill, G., et al. (2019) Mapping the world's free-flowing rivers. *Nature*, 569(7755), 215-221
- Guisan, A., Thuiller, W. and Zimmermann, N. E. (2017) *Habitat Suitability and Distribution Models (Cambridge University Press, Cambridge)*
- Hänfling, B., et al. (2016) Environmental DNA metabarcoding of lake fish communities reflects long-term data from established survey methods. *Molecular Ecology*, 25, 3101–3119.
- Kaelin, K., and Altermatt, F. (2016) Landscape-level predictions of diversity in river networks reveal opposing patterns for different groups of macroinvertebrates. *Aquatic Ecology* 50, 283-295.
- Laehnemann, D., Borkhardt, A., and McHardy, A. C. (2015) Denoising DNA deep sequencing data—high-throughput sequencing errors and their correction. *Briefings in bioinformatics* 17(1), 154–179.
- Leese, F., Sander, M., Buchner, D., Elbrecht, V., Haase, P. and Zizka, V. M.A. (2020) Improved freshwater macroinvertebrate detection from eDNA through minimized non-target amplification. *bioRxiv*.
- Leopold, L. B., and Maddock, T. (1953) *The hydraulic geometry of stream channels and some physiographic implications (Vol. 252)*. US Government Printing Office.
- Leopold, L. B., Wolman, M. G., and Miller, J. P. (1964) *Fluvial processes in geomorphology*. WH Freeman and Co. San Francisco, CA.
- Liu, C., Berry, P.M., Dawson, T.P. and Pearson, R.G. (2005) Selecting thresholds of occurrence in the prediction of species distributions. *Ecography*, 28: 385-393.
- Mächler, E. et al. (2019) Assessing different components of diversity across a river network using eDNA. *Environmental DNA*, 1, 290-301.
- Mächler, E., Walser, J. C., and Altermatt, F. (2020). Decision making and best practices for taxonomy-free eDNA metabarcoding in biomonitoring using Hill numbers. *bioRxiv*.

- O'Callaghan, J. F. and Mark, D. M. (1984) The extraction of drainage networks from digital elevation data. *Computer Vision, Graphics, and Image Processing*, 28, 323-344.
- Pawlowski J, et al. (2018) The future of biotic indices in the ecogenomic era: Integrating (e)DNA metabarcoding in biological assessment of aquatic ecosystems. *Science of The Total Environment*, 637–638: 1295-1310.
- Pawlowski, J., Apothéloz-Perret-Gentil, L., Mächler, E. and Altermatt, F. (2020). Environmental DNA applications in biomonitoring and bioassessment of aquatic ecosystems. Guidelines. Federal Office for the Environment, Bern. Environmental Studies. no. 2010. (in press).
- Pont, D., et al. (2018) Environmental DNA reveals quantitative patterns of fish biodiversity in large rivers despite its downstream transportation. *Scientific Reports*, 8(1).
- Rodriguez-Iturbe, I. and Rinaldo, A. (2001) *Fractal River Basins. Chance and self-organization*. Cambridge University Press.
- Shogren, A. J., Tank, J. L., Andruszkiewicz, E, Olds, B, Mahon, A. R., Jerde, C. L, and Bolster, D (2017) Controls on eDNA movement in streams: Transport, Retention, and Resuspension. *Scientific Reports* 7.
- Spens, J., et al. (2017) Comparison of capture and storage methods for aqueous microbial eDNA using an optimized extraction protocol: advantage of enclosed filter, *Methods in Ecology and Evolution* 8(5), 635–645.
- Thomsen, P. F., Møller, P. R., Sigsgaard, E. E., Knudsen, S. W., Jørgensen, O. A., and Willerslev, E. (2016) Environmental DNA from Seawater Samples Correlate with Trawl Catches of Subarctic, Deepwater Fishes. *PLoS ONE*, 11, e0165252.
- Ver Hoef, J. M., Peterson, E. E., Cliord, D., and Shah, R. (2014) SSN: An R package for spatial statistical modeling on stream networks. *Journal of Statistical Software*, 56(3), 1-45.
- Winter, D. J. (2017) rentrez: An R package for the NCBI eUtils API. *The R Journal* 9, 520-526.

Reviewers' Comments:

Reviewer #1:

Remarks to the Author:

Dear authors.

I have read though the rebuttal and I think that you have addressed and clarified my comments/suggestions appropriately.

Thus, I suggest that the paper is accepted.

Reviewer #2:

Remarks to the Author:

I have read the revised manuscript/supplement materials/updated figures/table and the responses to reviewers of this paper, titled "Environmental DNA allows upscaling spatial patterns of biodiversity in freshwater ecosystems." The authors have considered/addressed all the comments/feedback and given detailed explanations or reconduted methods/results for the concerns in the previous review. Specifically, the description/derivation of the hydrological principles underlying the model is to the satisfaction. The manuscript is now improved with greater clarity. I appreciate the authors' effort that made this interesting work more robust and is much more appreciated and understood by broader audiences. Thanks for the opportunity to provide comments and feedback.

*minor annotation attached.

Response to Reviewer #1

Dear authors. I have read though the rebuttal and I think that you have addressed and clarified my comments/suggestions appropriately. Thus, I suggest that the paper is accepted.

We wish to thank this Reviewer for the favorable assessment of our manuscript.

Response to Reviewer #2

I have read the revised manuscript/supplement materials/updated figures/table and the responses to reviewers of this paper, titled "Environmental DNA allows upscaling spatial patterns of biodiversity in freshwater ecosystems." The authors have considered/addressed all the comments/feedback and given detailed explanations or reconducted methods/results for the concerns in the previous review. Specifically, the description/derivation of the hydrological principles underlying the model is to the satisfaction. The manuscript is now improved with greater clarity. I appreciate the authors' effort that made this interesting work more robust and is much more appreciated and understood by broader audiences. Thanks for the opportunity to provide comments and feedback.

We wish to thank this Reviewer for the generous and extensive feedback provided. Below we provide replies to specific comments included in the attached PDF.

(L. 81) Kicknet show more species than eDNA detection?!

No. eDNA detection found 50 genera (see L. 77), while kicknet detection found 47 genera.

(L. 328) Same as Wang et al. 2011. Wang, L., Infante, D., Esselman, P., Cooper, A., Wu, D., Taylor, W., ... & Ostroff, A. (2011). A hierarchical spatial framework and database for the national river fish habitat condition assessment. Fisheries, 36(9), 436-449.

We wish to thank this reviewer for this interesting literature suggestion. However, we respectfully decided not to include it because it is not necessary (the method in question was not developed by Wang et al. in the first place), and because the number of references in the main text (74) is already slightly larger than the suggested limit of 70 for a Nature Communications article.